# EFFICIENT INVERSE MULTIAGENT LEARNING

**Denizalp Goktas**[*]**& Amy Greenwald**
Brown University, Computer Science
`denizalp_goktas@brown.edu`

**Sadie Zhao**
Harvard University, Computer Science

**Alex Koppel & Sumitra Ganesh**
JP Morgan Chase & Co

## ABSTRACT

In this paper, we study inverse game theory (resp. inverse multiagent learning) in which the goal is to find parameters of a game's payoff functions for which the expected (resp. sampled) behavior is an equilibrium. We formulate these problems as generative-adversarial (i.e., min-max) optimization problems, which we develop polynomial-time algorithms to solve, the former of which relies on an exact first-order oracle, and the latter, a stochastic one. We extend our approach to solve inverse multiagent simulacral learning in polynomial time and number of samples. In these problems, we seek a simulacrum, meaning parameters and an associated equilibrium that replicate the given observations in expectation. We find that our approach outperforms the widely-used ARIMA method in predicting prices in Spanish electricity markets based on time-series data.

## 1 INTRODUCTION

Game theory provides a mathematical framework, called *games*, which is used to predict the outcome of the interaction of preference-maximizing agents called *players*. Each player in a game chooses a *strategy* from its *strategy space* according to its preference relation, often represented by a *payoff function* over possible *outcomes*, implied by a *strategy profile* (i.e., a collection of strategies, one-per-player). The canonical outcome, or *solution concept*, prescribed by game theory is the *Nash equilibrium (NE)* (Nash, 1950): a strategy profile such that each player's strategy, fixing the equilibrium strategies of its opponents, is payoff-maximizing (or more generally, preference-maximizing).

In many applications of interest, such as contract design (Holmström, 1979; Grossman & Hart, 1992) and counterfactual prediction (Peysakhovich et al., 2019), the payoff functions (or more generally, preference relations) of the players are not available, but the players' strategies are. In such cases, we are concerned with estimating payoff functions for which these observed strategies are an equilibrium. This estimation task serves to *rationalize* the players' strategies (i.e., we can interpret the observed strategies as solutions to preference-maximization problems). Estimation problems of this nature characterize *inverse game theory* (Waugh et al., 2013; Bestick et al., 2013).

The primary object of study of inverse game theory is the *inverse game*, which comprises a game with the payoff functions omitted, and an *observed strategy profile*. The canonical solution concept prescribed for an inverse game is the *inverse Nash equilibrium*, i.e., payoff functions for which the observed strategy profile corresponds to a Nash equilibrium. If the set of payoff functions in an inverse game is unrestricted, the set of inverse Nash equilibria can contain a wide variety of spurious solutions, e.g., in all inverse games, the payoff function that assigns zero payoffs to all outcomes is an inverse Nash equilibrium, because any strategy profile is a Nash equilibrium of a *constant game*: i.e., one whose payoffs are constant across strategies. To meaningfully restrict the class of payoff functions over which to search for an inverse Nash equilibrium, one common approach (Kuleshov & Schrijvers, 2015; Syrgkanis et al., 2017) is to assume that the inverse game includes in addition to all the aforementioned objects, a *parameter-dependent payoff function* for each player, in which case an *inverse Nash equilibrium* is simply defined as parameter values such that the observed strategy profile is a Nash equilibrium of the parameter-dependent payoff functions evaluated at those values.

---

[*]Research conducted while the author was an intern at JP Morgan Chase & Co.

If one assumes *exact* oracle access to the payoffs of the game (i.e., if there exists a function which, for any strategy profile, returns the players' payoffs[1]), the problem of computing an inverse Nash equilibrium is one of *inverse multiagent planning*. In many games, however, a more appropriate assumption is *stochastic* oracle access, because of inherent stochasticity in the game (Shapley, 1953) or because players employ randomized strategies Nash (1950). The problem of computing an inverse Nash equilibrium assuming stochastic oracle access is one of *inverse multiagent learning*.

One important class of inverse games is that of *inverse Markov games*, in which the underlying game is a *Markov game* (Shapley, 1953; Fink, 1964; Takahashi, 1964), i.e., the game unfolds over an infinite time horizon: at each time period, players observe a state, take an action (simultaneously), receive a reward, and transition onto a new state. In such games, each player's strategy,[2] also called a *policy*, is a mapping from states to actions describing the action the player takes at each state, with any strategy profile inducing a *history distribution* over *histories of play* i.e., sequences of (state, action profile) tuples. The payoff for any strategy profile is then given by its *expected cumulative reward* over histories of play drawn from the history distribution associated with the strategy profile. Excluding rare instances,[3] the payoff function in Markov games is only accessible via a stochastic oracle, typically implemented via a game simulator that returns *estimates* of the value of the game's rewards and transition probabilities. As such, the computation of an inverse Nash equilibrium in an inverse Markov game is an inverse multiagent learning problem, which is often called *inverse multiagent reinforcement learning (inverse MARL)* (Natarajan et al., 2010).

In many real-world applications of inverse Markov games, such as robotics control (Coates et al., 2009), one does not directly observe Nash equilibrium strategies but rather histories of play, which we assume are sampled from the history distribution associated with some Nash equilibrium. In these applications, we are given *an inverse simulation*—an inverse Markov game together with sample histories of play—based on which we seek parameter values which induce payoff functions that rationalize the observed histories. As a Nash equilibrium itself is not directly observed in this setting, we aim to compute parameter values that induce a Nash equilibrium that replicates the observed histories *in expectation*. We call the solution of such an inverse simulation (i.e., parameter values together with an associated Nash equilibrium) a *simulacrum*. Not only does a simulacrum serve to explain (i.e., rationalize) observations, additionally, it can provide predictions of unobserved behavior.

We study two *simulacral learning* problems, a first-order version in which samples histories of play are faithful, and a second-order version in which they are not—a (possibly stochastic) function of each history of play is observed rather than the history itself. Here, the use of the term "first-order" refers to the fact that the simulacrum does not necessarily imitate the actual equilibrium that generated the histories of play, since multiple equilibria can generate the same histories of play (Baudrillard, 1994). More generally, if the simulacrum is "second-order," it is nonfaithful, meaning some information about the sample histories of play is lost. We refer to the problems of computing first-order (resp. second-order) simulacra as *first-order (resp. second-order) simulacral learning*: i.e., build a first-order (resp. second-order) simulacrum from faithful (resp. non-faithful; e.g., aggregate agent behavior) sample histories of play. We summarize the problems characterizing inverse game theory in Table 1a.

**Contributions**  The algorithms introduced in this paper extend the class of games for which an inverse Nash equilibrium can be computed efficiently (i.e., in polynomial-time) to the class of normal-form concave games (which includes normal-form finite action games), finite state and action Markov games, and a large class of continuous state and action Markov games. While our focus is on Markov games in this paper, the results apply to normal-form (Nash, 1950), Bayesian (Harsanyi, 1967; 1968), and extensive-form games (Zermelo, 1913). The results also extend to other equilibrium concepts, beyond Nash, such as (coarse) correlated Aumann (1974); Moulin & Vial (1978), and more generally, $\Phi$-equilibrium (Greenwald & Jafari, 2003) *mutatis mutandis*.

First, regarding inverse multiagent planning, we provide a min-max characterization of the set of inverse Nash equilibria of any inverse game for which the set of inverse Nash equilibria is non-empty, assuming an exact oracle (Theorem 3.1). We then show that for any inverse concave game, when the

---

[1] Throughout this work, we assume that the oracle evaluations are constant time and measure computational complexity in terms of the number of oracle calls.

[2] Throughout this paper, a strategy refers to the complete description of a players' behavior at any state or time of the game, while an action refers to a specific realization of a strategy at a given state and time.

[3] For simple enough games, one can express the expected cumulative reward in closed form, and then solve the inverse (stochastic) game assuming exact oracle access.

| Equilibrium Access | Exact Oracle | Stochastic Oracle |
|---|---|---|
| Direct | Inverse Multiagent Planning | Inverse Multiagent Learning |
| Faithful Samples | First-order Simulacral Planning | First-order Simulacral Learning |
| Nonfaithful Samples | Second-order Simulacral Planning | Second-order Simulacral Learning |

(a) Taxonomy of inverse game theory problems. First-order simulacral learning is more commonly known as multiagent apprenticeship learning (Abbeel & Ng, 2004; Yang et al., 2020).

| Reference | Game Type | Solution Concept | Polytime? |
|---|---|---|---|
| (Fu et al., 2021) | Finite Markov | Nash | ✗ |
| (Yu et al., 2019) | Finite Markov | Quantal Response | ✗ |
| (Lin et al., 2019) | Finite Zero-sum Markov | Various | ✗ |
| (Song et al., 2018) | Finite Markov | Quantal Response | ✗ |
| (Syrgkanis et al., 2017) | Finite Bayesian | Bayes-Nash | ✓ |
| (Kuleshov & Schrijvers, 2015) | Finite Normal-Form | Correlated | ✓ |
| (Waugh et al., 2013) | Finite Normal-Form | Correlated | ✓ |
| (Bestick et al., 2013) | Finite Normal-Form | Correlated | ✗ |
| (Natarajan et al., 2010) | Finite Markov | Cooperative | ✗ |
| This work | Finite/Concave Normal-form Finite/Concave Markov | Nash/Correlated Any Other | ✓ |

(b) A comparison of our work and prior work on inverse game theory and inverse MARL.

regret of each player is convex in the parameters of the inverse game, an assumption satisfied by a large class of inverse games such as inverse normal-form games, this min-max optimization problem is convex-concave, and can thus be solved in polynomial time (Theorem 3.2) via standard first-order methods. This characterization also shows that the set of inverse Nash equilibria can be convex, even when the set of Nash equilibria is not.

Second, we generalize our min-max characterization to inverse multiagent learning, in particular inverse MARL, where we are given an inverse Markov game, and correspondingly, a *stochastic* oracle, and we seek a first-order simulacrum (Corollary 1). We show that under standard assumptions, which are satisfied by a large class of inverse Markov games (e.g., all finite state and action Markov games and a class of continuous state and action Markov games), the ensuing min-max optimization problem is convex-gradient dominated, and thus an inverse Nash equilibrium can be computed once again via standard first-order methods in polynomial time (Theorem 4.1).

Third, we provide an extension of our min-max characterization to (second-order) simulacral learning (Theorem 5.1). We once again characterize the problem as a solution to a min-max optimization problem, for which standard first-order methods compute a first-order stationary (Lin et al., 2020) solution in polynomial-time, using a number of observations (i.e., unfaithful samples of histories of play) that is polynomial in the size of the inverse simulation (Theorem 5.2).

Finally, we include two sets of experiments. In the first, we show that our method is effective in synthetic economic settings where the goal is to recover buyers' valuations from observed competitive equilibria (which, in this market, coincide with Nash equilibria). Second, using real-world time-series data, we apply our method to predict prices in Spanish electricity markets, and find that it outperforms the widely-used ARIMA method in predicting prices on this real-world data set.

## 2 PRELIMINARIES

**Notation.** All notation for variable types, e.g., vectors, should be clear from context; if any confusion arises, see Section 7.1. We denote by $[n]$ the set of integers $\{1, \ldots, n\}$. Let $\mathcal{X}$ be any set and $(\mathcal{X}, \mathcal{F})$ any associated measurable space, where the $\sigma$-algebra $\mathcal{F}$ unless otherwise noted is assumed to be the $\sigma$-algebra of countable sets, i.e., $\mathcal{F} \doteq \{\mathcal{E} \subseteq \mathcal{X} \mid \mathcal{E} \text{ is countable }\}$. We write $\Delta(\mathcal{X}) \doteq \{\mu : (\mathcal{X}, \mathcal{F}) \to [0, 1]\}$ to denote the set of *probability measures* on $(\mathcal{X}, \mathcal{F})$. Additionally, we denote the orthogonal projection operator onto a set $\mathcal{X}$ by $\Pi_{\mathcal{X}}(\boldsymbol{x}) \doteq \arg\min_{\boldsymbol{y} \in \mathcal{X}} \|\boldsymbol{x} - \boldsymbol{y}\|_2^2$.

**Mathematical Concepts.** Consider any normed space $(\mathcal{X}, \|\cdot\|)$ where $\mathcal{X} \subset \mathbb{R}^m$ and any function $f : \mathcal{X} \to \mathbb{R}$. $f$ is $\ell_f$-*Lipschitz-continuous* w.r.t. norm (typically, Euclidean) $\|\cdot\|$ iff $\forall \boldsymbol{x}_1, \boldsymbol{x}_2 \in \mathcal{A}, \|f(\boldsymbol{x}_1) - f(\boldsymbol{x}_2)\| \le \ell_f \|\boldsymbol{x}_1 - \boldsymbol{x}_2\|$. If the gradient of $f$ is $\ell_{\nabla f}$-Lipschitz-continuous, we refer to $f$ as $\ell_{\nabla f}$-*Lipschitz-smooth*. Furthermore, given $\mu > 0$, $f$ is said to be $\mu$-*gradient-dominated* if $\min_{\boldsymbol{x}' \in \mathcal{X}} f(\boldsymbol{x}') \ge f(\boldsymbol{x}) + \mu \cdot \min_{\boldsymbol{x}' \in \mathcal{X}} \langle \boldsymbol{x}' - \boldsymbol{x}, \nabla f(\boldsymbol{x}) \rangle$ (Bhandari & Russo, 2019).

**Normal-form Games.** A *(parametric) game* $\mathcal{G}^{\boldsymbol{\theta}} \doteq (n, m, d, \mathcal{X}, \Theta, \boldsymbol{\theta}, \boldsymbol{u})$ comprises $n \in \mathbb{N}_+$ players, each $i \in [n]$ of whom chooses a strategy $\boldsymbol{x}_i \in \mathcal{X}_i$ from an strategy space $\mathcal{X}_i \subseteq \mathbb{R}^m$ simultaneously. We refer to any vector of per-player strategies $\boldsymbol{x} = (\boldsymbol{x}_1, \ldots, \boldsymbol{x}_n) \in \mathcal{X}$ as a *strategy profile*, where $\mathcal{X} \doteq \bigtimes_{i \in [n]} \mathcal{X}_i \subseteq \mathbb{R}^{nm}$ denotes the space of all strategy profiles. After the players choose their strategies $\boldsymbol{x} \in \mathcal{X}$, each receives a payoff $u_i(\boldsymbol{x}; \boldsymbol{\theta})$ given by payoff function $u_i : \mathcal{X} \times \Theta \to \mathbb{R}$ parameterized by a vector $\boldsymbol{\theta}$ in a parameter space $\Theta \subseteq \mathbb{R}^d$. We define the *payoff profile function* $\boldsymbol{u}(\boldsymbol{x}; \boldsymbol{\theta}) \doteq (u_i(\boldsymbol{x}; \boldsymbol{\theta}))_{i \in [n]}$; the *cumulative regret* $\psi : \mathcal{X} \times \mathcal{X} \times \Theta \to \mathbb{R}$ across all players, between two strategy profiles $\boldsymbol{x}, \boldsymbol{y} \in \mathcal{X}$, given $\boldsymbol{\theta} \in \Theta$, as $\psi(\boldsymbol{x}, \boldsymbol{y}; \boldsymbol{\theta}) \doteq \sum_{i \in [n]} u_i(\boldsymbol{y}_i, \boldsymbol{x}_{-i}; \boldsymbol{\theta}) - u_i(\boldsymbol{x}; \boldsymbol{\theta})$; and the *exploitability* (or *Nikaido-Isoda potential* (Nikaido & Isoda, 1955)) $\varphi(\boldsymbol{x}; \boldsymbol{\theta}) \doteq \max_{\boldsymbol{y} \in \mathcal{X}} \psi(\boldsymbol{x}, \boldsymbol{y}; \boldsymbol{\theta})$.

A game is said to be *concave* if for all parameters $\boldsymbol{\theta} \in \Theta$ and players $i \in [n]$, 1. $\mathcal{X}_i$ is non-empty, compact, and convex, 2. $u_i$ is continuous, and 3. $\boldsymbol{x}_i \mapsto u_i(\boldsymbol{x}_i, \boldsymbol{x}_{-i}; \boldsymbol{\theta})$ is concave. Given $\boldsymbol{\theta} \in \Theta$, an *$\varepsilon$-Nash equilibrium* ($\varepsilon$-NE) of a game $\mathcal{G}^{\boldsymbol{\theta}}$ is a strategy profile $\boldsymbol{x}^* \in \mathcal{X}$ s.t. $u_i(\boldsymbol{x}^*; \boldsymbol{\theta}) \geq \max_{\boldsymbol{x}_i \in \mathcal{X}_i} u_i(\boldsymbol{x}_i, \boldsymbol{x}^*_{-i}; \boldsymbol{\theta}) - \varepsilon$, for all players $i \in [n]$. A 0-Nash equilibrium is simply called a Nash equilibrium, and is guaranteed to exist in concave games (Nash, 1950; Arrow & Debreu, 1954).

**Dynamic Games.** An *(infinite-horizon, discounted, parametric) Markov game* (Shapley, 1953; Fink, 1964; Takahashi, 1964) $\mathcal{M}^{\boldsymbol{\theta}} \doteq (n, m, \mathcal{S}, \mathcal{A}, \Theta, \boldsymbol{\theta}, \boldsymbol{r}, p, \gamma, \mu)$ is a dynamic game played over an infinite time horizon. The game initiates at time $t = 0$ in some state $S^{(0)} \sim \mu$ drawn from an *initial state distribution* $\mu \in \Delta(\mathcal{S})$. At each time period $t = 0, 1, \ldots$, each player $i \in [n]$ plays an *action* $\boldsymbol{a}_i^{(t)} \in \mathcal{A}_i$ from an action space $\mathcal{A}_i \subset \mathbb{R}^m$. We define the space of action profiles $\mathcal{A} = \times_{i \in [n]} \mathcal{A}_i$. After the players choose their *action profile* $\boldsymbol{a}^{(t)} \doteq (\boldsymbol{a}_1^{(t)}, \ldots, \boldsymbol{a}_n^{(t)}) \in \mathcal{A}$, each player $i$ receives a *reward* $r_i(\boldsymbol{s}^{(t)}, \boldsymbol{a}^{(t)}; \boldsymbol{\theta})$ according to a *parameterized reward profile function* $\boldsymbol{r} : \mathcal{S} \times \mathcal{A} \times \Theta \to \mathbb{R}^n$. The game then either ends with probability $1 - \gamma$, where $\gamma \in (0, 1)$ is called the *discount factor*, or transitions to a new state $S^{(t+1)} \sim p(\cdot \mid \boldsymbol{s}^{(t)}, \boldsymbol{a}^{(t)})$ according to a *(Markov) probability transition kernel* $p$ whereby for all $(\boldsymbol{s}, \boldsymbol{a}) \in \mathcal{S} \times \mathcal{A}$, $p(\cdot \mid \boldsymbol{s}, \boldsymbol{a}) \in \Delta(\mathcal{S})$, and $p(\boldsymbol{s}^{(t+1)} \mid \boldsymbol{s}^{(t)}, \boldsymbol{a}^{(t)}) = \mathbb{P}(S^{(t+1)} = \boldsymbol{s}^{(t+1)} \mid S^{(t)} = \boldsymbol{s}^{(t)}, A^{(t)} = \boldsymbol{a}^{(t)})$ is the probability of transitioning to state $\boldsymbol{s}^{(t+1)}$ from state $\boldsymbol{s}^{(t)}$ when the players' take action profile $\boldsymbol{a}^{(t)}$.[4]

A *(stationary Markov) policy* (Maskin & Tirole, 2001) for player $i \in [n]$ is a mapping $\boldsymbol{\pi}_i : \mathcal{S} \to \mathcal{A}$ from states to actions so that $\boldsymbol{\pi}_i(\boldsymbol{s}) \in \mathcal{A}_i$ denotes the action that player $i$ takes at state $\boldsymbol{s}$. For each player $i \in [n]$, we define the space of all (measurable) policies $\mathcal{P}_i \doteq \{\boldsymbol{\pi}_i : \mathcal{S} \to \mathcal{A}_i\}$. As usual, $\boldsymbol{\pi} \doteq (\boldsymbol{\pi}_1, \ldots, \boldsymbol{\pi}_n) \in \mathcal{P} \doteq \times_{i \in [n]} \mathcal{P}_i$ denotes a *policy profile*. A *history (of play)* $\boldsymbol{h} \in (\mathcal{S} \times \mathcal{A})^T$ of length $T \in \mathbb{N}$ is a sequence of state-action tuples $\boldsymbol{h} = (\boldsymbol{s}^{(t)}, \boldsymbol{a}^{(t)})_{t=0}^{T-1}$. For any policy profile $\boldsymbol{\pi} \in \mathcal{P}$, define the *discounted history distribution* $\nu^{\boldsymbol{\pi}}(\boldsymbol{h}) \doteq \mu(\boldsymbol{s}^{(0)}) \prod_{t=0}^{T} \gamma^t p(\boldsymbol{s}^{(t+1)} \mid \boldsymbol{s}^{(t)}, \boldsymbol{\pi}(\boldsymbol{s}^{(t)}))$ as the probability of observing a history $\boldsymbol{h}$ of length $T$. Throughout, we denote by $H \doteq (S^{(t)}, A^{(t)})_t \sim \nu^{\boldsymbol{\pi}}$ any randomly sampled history from $\nu^{\boldsymbol{\pi}}$.[5]

Fix a policy profile $\boldsymbol{\pi} \in \mathcal{P}$ and a player $i$. In our analysis of Markov games, we rely on the following terminology. The *expected cumulative payoff* is given by $u_i(\boldsymbol{\pi}; \boldsymbol{\theta}) \doteq \mathbb{E}_{H \sim \nu^{\boldsymbol{\pi}}} \left[ \sum_{t=0}^{\infty} r_i(S^{(t)}, A^{(t)}; \boldsymbol{\theta}) \right]$. The *state-* and *action-value functions* are defined, respectively, as $v_i^{\boldsymbol{\pi}}(\boldsymbol{s}; \boldsymbol{\theta}) \doteq \mathbb{E}_{H \sim \nu^{\boldsymbol{\pi}}} \left[ \sum_{t=0}^{\infty} r_i(S^{(t)}, A^{(t)}; \boldsymbol{\theta}) \mid S^{(0)} = \boldsymbol{s} \right]$ and $q_i^{\boldsymbol{\pi}}(\boldsymbol{s}, \boldsymbol{a}; \boldsymbol{\theta}) \doteq \mathbb{E}_{H \sim \nu^{\boldsymbol{\pi}}} \left[ \sum_{t=0}^{\infty} r_i(S^{(t)}, A^{(t)}; \boldsymbol{\theta}) \mid S^{(0)} = \boldsymbol{s}, A^{(0)} = \boldsymbol{a} \right]$. The *state occupancy distribution* $\delta_\mu^{\boldsymbol{\pi}} \in \Delta(\mathcal{S})$ denotes the probability that a state is reached under a policy $\boldsymbol{\pi}$, given initial state distribution $\mu$, i.e., $\delta_\mu^{\boldsymbol{\pi}}(\boldsymbol{s}) \doteq \mathbb{E}_{H \sim \nu^{\boldsymbol{\pi}}} \left[ \sum_{t=0}^{\infty} \mathbb{1}_{S^{(t)} = \boldsymbol{s}} \right]$. Finally, as usual, an *$\varepsilon$-Nash equilibrium* ($\varepsilon$-NE) of a game $\mathcal{M}^{\boldsymbol{\theta}}$ is a policy profile $\boldsymbol{\pi}^* \in \mathcal{P}$ such that for all $i \in [n]$, $u_i(\boldsymbol{\pi}^*; \boldsymbol{\theta}) \geq \max_{\boldsymbol{\pi}_i \in \mathcal{P}_i} u_i(\boldsymbol{\pi}_i, \boldsymbol{\pi}^*_{-i}; \boldsymbol{\theta}) - \varepsilon$; and a Nash equilibrium ensues when $\varepsilon = 0$.

## 3 INVERSE MULTIAGENT PLANNING

The goal of inverse multiagent planning is to invert an equilibrium: i.e., estimate a game's parameters, given observed behavior. In this section, we present our main idea, namely a zero-sum game (i.e., min-max optimization) characterization of inverse multiagent planning, where one player called

---

[4]For notational convenience, we assume the probability transition function is independent of the parameters, but we note that our min-max characterizations apply more broadly without any additional assumptions, while our polynomial-time computation results apply when, in addition to Assumption 4, one assumes the probability transition function is stochastically convex (see, for instance, Atakan (2003a)) in the parameters of the game.

[5]Let $(\mathcal{S}, \mathcal{F}_{\mathcal{S}})$, $(\mathcal{A}, \mathcal{F}_{\mathcal{A}})$, and $(\mathcal{S} \times \mathcal{A}, \mathcal{F}_{\mathcal{S} \times \mathcal{A}})$ be the measurable spaces associated with the state, action profile, and state-action profile ($\mathcal{S} \times \mathcal{A}$) spaces, respectively. Further, let $([0, 1], \mathcal{B}_{[0,1]})$, $(\mathbb{R}^n, \mathcal{B}_{\mathbb{R}^n})$ be measurable spaces on $[0, 1]$ and $\mathbb{R}^n$ defined by the Borel $\sigma$-algebra. For simplicity, we do not explicitly represent the reward profile function, transition probability kernel, initial state distribution, or policies as measures or measurable functions. We note, however, that for the expectations we define to be well-posed, they all must be assumed to be measurable functions. We simply write $\boldsymbol{r} : \mathcal{S} \times \mathcal{A} \to \mathbb{R}^n$, $p : \mathcal{S} \times (\mathcal{S} \times \mathcal{A}) \to [0, 1]$, $\mu : \mathcal{S} \to [0, 1]$, and $\boldsymbol{\pi} : \mathcal{S} \to \mathcal{A}$ to mean, respectively, $\boldsymbol{r} : (\mathcal{S} \times \mathcal{A}, \mathcal{F}_{\mathcal{S} \times \mathcal{A}}) \to (\mathbb{R}^n, \mathcal{B}_{\mathbb{R}^n})$, $p : (\mathcal{S}, \mathcal{F}_{\mathcal{S}}) \times (\mathcal{S} \times \mathcal{A}, \mathcal{F}_{\mathcal{S} \times \mathcal{A}}) \to ([0, 1], \mathcal{B}_{[0,1]})$, $\mu : (\mathcal{S}, \mathcal{F}_{\mathcal{S}}) \to ([0, 1], \mathcal{B}_{[0,1]})$, and $\boldsymbol{\pi} : (\mathcal{S}, \mathcal{F}_{\mathcal{S}}) \to (\mathcal{A}, \mathcal{F}_{\mathcal{A}})$.

the stabilizer picks parameters, while the other called the destabilizer picks per-player deviations. This game is zero-sum) because the stabilizer seeks parameters that rationalize (i.e., minimize the exploitability of) the observed equilibrium, while the destabilizer aims to rebut the rationality of the observed equilibrium (i.e., seeks deviations that maximize cumulative regret). We use this characterization to develop a gradient descent ascent algorithm that finds inverse NE in polynomial time, assuming access to *an exact first-order oracle*: specifically, a pair of functions that return the value and gradient of the payoff profile function.

An *inverse game* $\mathcal{G}^{-1} \doteq (\mathcal{G}^{\boldsymbol{\theta}^\dagger} \setminus \boldsymbol{\theta}^\dagger, \boldsymbol{x}^\dagger)$ comprises a *game form* (i.e., a parametric game *sans* its parameter) $\mathcal{G}^{\boldsymbol{\theta}^\dagger} \setminus \boldsymbol{\theta}^\dagger$ together with an observed strategy profile $\boldsymbol{x}^\dagger$, which we assume is a Nash equilibrium. Crucially, we do not observe the parameters $\boldsymbol{\theta}^\dagger$ of the payoff functions. Given an inverse game $\mathcal{G}^{-1}$, our goal is to compute an $\varepsilon$-*inverse Nash equilibrium*, meaning parameter values $\boldsymbol{\theta}^* \in \Theta$ s.t. $\boldsymbol{x}^\dagger \in \mathcal{X}$ is an $\varepsilon$-NE of $\mathcal{G}^{\boldsymbol{\theta}^*}$. As usual, a 0-inverse NE is simply called an inverse NE. Note that this definition does not require that we identify the true parameters $\boldsymbol{\theta}^\dagger$, as identifying $\boldsymbol{\theta}^\dagger$ is impossible unless there exists a bijection between the set of parameters and the set of Nash equilibria, a highly restrictive assumption that is not even satisfied in games with a unique Nash equilibrium. To compute an inverse NE is to find parameter values that minimize the exploitability of the observed equilibrium. This problem is a min-max optimization problem, as the parameter values that minimize exploitability are those that maximize the players' cumulative regrets. More precisely:

**Theorem 3.1.** *The set of inverse NE of $\mathcal{G}^{-1}$ is the set of parameter profiles $\boldsymbol{\theta} \in \Theta$ that solve the optimization problem $\min_{\boldsymbol{\theta} \in \Theta} \varphi(\boldsymbol{x}^\dagger; \boldsymbol{\theta})$, or equivalently, this min-max optimization problem:*

$$\min_{\boldsymbol{\theta} \in \Theta} \max_{\boldsymbol{y} \in \mathcal{X}} f(\boldsymbol{\theta}, \boldsymbol{y}) \doteq \psi(\boldsymbol{x}^\dagger, \boldsymbol{y}; \boldsymbol{\theta}) = \sum_{i \in [n]} \left[ u_i(\boldsymbol{y}_i, \boldsymbol{x}^\dagger_{-i}; \boldsymbol{\theta}) - u_i(\boldsymbol{x}^\dagger; \boldsymbol{\theta}) \right] \quad (1)$$

This min-max optimization problem can be seen as a generalization of the dual of Waugh et al.'s (2013) maximum entropy likelihood maximization method for games with possibly continuous strategy spaces, taking Nash equilibrium rather than maximum entropy correlated equilibrium as the inverse equilibrium. In contrast to Waugh et al.'s dual, our min-max optimization problem characterizes the set of *all* inverse NE, and not only a subset of the inverse correlated equilibria, in particular those that maximize entropy. This formulation also generalizes Swamy et al.'s (2021) moment matching game from a single-agent to a multiagent setting.

---

**Algorithm 1** Adversarial Inverse Multiagent Planning

**Inputs:** $\Theta, \mathcal{X}, f, \boldsymbol{\eta_\theta}, \boldsymbol{\eta_y}, T, \boldsymbol{\theta}^{(0)}, \boldsymbol{y}^{(0)}, \boldsymbol{x}^\dagger$
**Outputs:** $(\boldsymbol{\theta}^{(t)}, \boldsymbol{y}^{(t)})_{t=0}^T$
1: **for** $t = 0, \ldots, T-1$ **do**
2: $\quad \boldsymbol{\theta}^{(t+1)} \leftarrow \Pi_\Theta \left[ \boldsymbol{\theta}^{(t)} - \eta_{\boldsymbol{\theta}}^{(t)} \nabla_{\boldsymbol{\theta}} f(\boldsymbol{\theta}^{(t)}, \boldsymbol{y}^{(t)}) \right]$
3: $\quad \boldsymbol{y}^{(t+1)} \leftarrow \Pi_\mathcal{X} \left[ \boldsymbol{y}^{(t)} + \eta_{\boldsymbol{y}}^{(t)} \nabla_{\boldsymbol{y}} f(\boldsymbol{\theta}^{(t)}, \boldsymbol{y}^{(t)}) \right]$
4: **return** $(\boldsymbol{\theta}^{(t)}, \boldsymbol{y}^{(t)})_{t=0}^T$

---

Without further assumptions, the objective function $f$ in Equation (1) is non-convex non-concave; however, under suitable assumptions (Assumption 1) satisfied by finite action normal-form games, for example, it becomes convex-concave.

**Assumption 1.** *Given an inverse game $\mathcal{G}^{-1}$, assume 1. (Concave game) for all parameters $\boldsymbol{\theta} \in \Theta$, $\mathcal{G}^{\boldsymbol{\theta}}$ is concave; and 2. (Convex parametrization) $\Theta$ is non-empty, compact, and convex; and for all $\forall i \in [n]$, $\boldsymbol{y}_i \in \mathcal{X}_i$, and $\boldsymbol{x}^\dagger \in \mathcal{X}$, each player $i$'s regret $\boldsymbol{\theta} \mapsto u_i(\boldsymbol{y}_i, \boldsymbol{x}^\dagger_{-i}; \boldsymbol{\theta}) - u_i(\boldsymbol{x}^\dagger; \boldsymbol{\theta})$ is convex.*

**Remark 1.** *Perhaps surprisingly, the set of inverse NE can be convex even when the set of NE is not, since the set of solutions to a convex-concave (or even convex-non-concave) min-max optimization problem is convex. This observation should alleviate any worries about the computational intractability of inverse game theory that might have been suggested by the computational intractability of game theory itself (Daskalakis et al., 2009; Chen & Deng, 2006).*

If additionally, we assume the players' payoffs are Lipschitz-smooth (Assumption 2), Equation (1) can then be solved to $\varepsilon$ precision in $O(1/\varepsilon^2)$ via gradient descent ascent (Algorithm 1). That is, as Theorem 3.2 shows, an inverse $\varepsilon$-NE can be computed in $O(1/\varepsilon^2)$ iterations.[6] We note that this convergence complexity can be further reduced to $O(1/\varepsilon)$ (even without decreasing step-sizes) if one instead applies an extragradient descent ascent method (Golowich et al., 2020) or optimistic GDA (Gorbunov et al., 2022).

---

[6]We include detailed theorem statements and proofs in Section 7.2.

**Assumption 2** (Lipschitz-Smooth Game). *For all players $i \in [n]$, $u_i$ is $\ell_{\nabla u_i}$-Lipschitz-smooth.*

**Theorem 3.2** (Inverse NE Complexity). *Under Assumptions 1–2, for $\varepsilon \geq 0$, if Algorithm 1 is run with inputs that satisfy $T \in \Omega(1/\varepsilon^2)$ and for all $t \in [T]$, $\eta_{\boldsymbol{y}}^{(t)} = \eta_{\boldsymbol{\theta}}^{(t)} \asymp 1/t$, then the time-average of all parameters $\overline{\boldsymbol{\theta}^{(T)}} \doteq \frac{1}{T+1}\sum_{t=0}^{T}\boldsymbol{\theta}^{(t)}$ is an $\varepsilon$-inverse NE.*

# 4 INVERSE MULTIAGENT REINFORCEMENT LEARNING

In this section, we build on our zero-sum game (i.e., min-max optimization) characterization of inverse game theory to tackle inverse MARL in an analogous fashion. As it is unreasonable to assume exact oracle access to the players' (cumulative) payoffs in inverse MARL, we relax this assumption in favor of a stochastic oracle model. More specifically, we assume access to a *differentiable game simulator* (Suh et al., 2022), which simulates histories of play $h \sim \nu^{\boldsymbol{\pi}}$ according to $\nu^{\boldsymbol{\pi}}$, given any policy profile $\boldsymbol{\pi}$, and returns the rewards $\boldsymbol{r}$ and transition probabilities $p,$[7] encountered along the way, together with their gradients.

Formally, an *inverse Markov game* $\mathcal{M}^{-1} \doteq (\mathcal{M}^{\boldsymbol{\theta}^\dagger} \setminus \boldsymbol{\theta}^\dagger, \boldsymbol{\pi}^\dagger)$ is an inverse game that comprises a *Markov game form* (i.e., a parametric Markov game *sans* its parameter) $\mathcal{M}^{\boldsymbol{\theta}^\dagger} \setminus \boldsymbol{\theta}^\dagger$ together with an observed policy profile $\boldsymbol{\pi}^\dagger$, which we assume is a Nash equilibrium. Crucially, we do not observe the parameters $\boldsymbol{\theta}^\dagger$ of the payoff functions. Since a Markov game is a normal-form game with payoffs given by $\boldsymbol{u}(\boldsymbol{\pi}; \boldsymbol{\theta}) = \mathbb{E}_{H \sim \nu^{\boldsymbol{\pi}}}\left[\sum_{t=0}^{\infty}\boldsymbol{r}(S^{(t)}, A^{(t)}; \boldsymbol{\theta})\right]$, the usual definitions of inverse NE and cumulative regret apply, and the following result, which characterizes the set of inverse NE as the minimizers of a *stochastic* min-max optimization problem, is a corollary of Theorem 3.1.

**Corollary 1.** *The set of inverse NE of $\mathcal{M}^{-1}$ is characterized by solutions to the following problem:*

$$\min_{\boldsymbol{\theta} \in \Theta} \max_{\boldsymbol{\pi} \in \mathcal{P}} f(\boldsymbol{\theta}, \boldsymbol{\pi}) \doteq \sum_{i \in [n]} \mathop{\mathbb{E}}_{\substack{H \sim \nu^{(\boldsymbol{\pi}_i, \boldsymbol{\pi}^\dagger_{-i})} \\ H' \sim \nu^{\boldsymbol{\pi}^\dagger}}} \left[\sum_{t=0}^{\infty} r_i(S^{(t)}, A^{(t)}; \boldsymbol{\theta}) - \sum_{t=0}^{\infty} r_i(S^{\dagger(t)}, A^{\dagger(t)}; \boldsymbol{\theta})\right] \quad (2)$$

As is usual in reinforcement learning, we use policy gradient to solve the destabilizer's problem in Equation (2). To do so, we restrict the destabilizer's action space to a policy class $\mathcal{P}^{\mathcal{X}}$ parameterized by $\mathcal{X} \subset \mathbb{R}^l$. Redefining $f(\boldsymbol{\theta}, \boldsymbol{x}) \doteq f(\boldsymbol{\theta}, \boldsymbol{\pi^x})$, for $\pi_{\boldsymbol{x}} \in \mathcal{P}^{\mathcal{X}}$, we aim to solve the stochastic min-max optimization problem $\min_{\boldsymbol{\theta} \in \Theta} \max_{\boldsymbol{x} \in \mathcal{X}} f(\boldsymbol{\theta}, \boldsymbol{x})$. Solutions to this problem are a superset of the solutions to Equation (2), unless it so happens that all best responses can be represented by policies in $\mathcal{P}^{\mathcal{X}}$, because restricting the expressivity of the policy class decreases the power of the destabilizer. As in Section 3, without any additional assumptions, $f$ is in general non-convex, non-concave, and non-smooth. While we can ensure convexity and smoothness of $\boldsymbol{\theta} \mapsto f(\boldsymbol{\theta}, \boldsymbol{x})$ under suitable assumptions on the game parameterization, namely by assuming the regret at each state is convex in $\boldsymbol{\theta}$, concavity in $\boldsymbol{x}$ is not satisfied even by finite state and action Markov games. Under the following conditions, however, we can guarantee that $f$ is Lipschitz-smooth, convex in $\boldsymbol{\theta}$, and gradient dominated in $\boldsymbol{x}$.

**Assumption 3** (Lipschitz-Smooth Gradient-Dominated Game). *Given an inverse Markov game $\mathcal{M}^{-1}$, assume 1. $\mathcal{S}$ and $\mathcal{A}$ are non-empty, and compact; 2. (Convex parameter spaces) $\mathcal{X}, \Theta$ are non-empty, compact, and convex; 3. (Smooth Game) $\nabla \boldsymbol{r}$, $\nabla p$, and $\nabla_{\boldsymbol{x}} \boldsymbol{\pi^x}$, for all policies $\boldsymbol{\pi^x} \in \mathcal{P}^{\mathcal{X}}$, are continuously differentiable; 4. (Gradient-Dominated Game) for all players $i \in [n]$, states $\boldsymbol{s} \in \mathcal{S}$, action profiles $\boldsymbol{a} \in \mathcal{A}$, and policies $\boldsymbol{\pi^x} \in \mathcal{P}^{\mathcal{X}}$, $\boldsymbol{x} \mapsto q_i^{\boldsymbol{\pi^x}}(\boldsymbol{s}, \boldsymbol{\pi^x}(\boldsymbol{s}); \boldsymbol{\theta})$ is $\mu$-gradient-dominated for some $\mu > 0$; and 5. (Closure under Policy Improvement) for all states $\boldsymbol{s} \in \mathcal{S}$, players $i \in [n]$, and policy profiles $\boldsymbol{\pi} \in \mathcal{P}$, there exists $\boldsymbol{\pi^x} \in \mathcal{P}^{\mathcal{X}}$ s.t. $q_i^{\boldsymbol{\pi}}(\boldsymbol{s}, \boldsymbol{\pi}_i^{\boldsymbol{x}}(\boldsymbol{s}), \boldsymbol{\pi}_{-i}(\boldsymbol{s})) = \max_{\boldsymbol{\pi}_i' \in \mathcal{P}_i} q_i^{\boldsymbol{\pi}}(\boldsymbol{s}, \boldsymbol{\pi}_i'(\boldsymbol{s}), \boldsymbol{\pi}_{-i}(\boldsymbol{s}))$.*

Part 3 of Assumption 3 implies that the game's cumulative payoff function is Lipschitz-smooth in the policy parameters $\boldsymbol{x}$. We note that a large class of Markov games satisfy Part 4, including linear quadratic games (Bhandari & Russo, 2019), finite state and action games, and continuous state and action games whose rewards (resp. transition probabilities) are concave (resp. stochastically

---

[7]We note that in inverse reinforcement learning, as opposed to reinforcement learning, it is typical to assume that the transition model is known (see, for instance (Abbeel & Ng, 2004), Footnote 8).

concave) in each player's action (Atakan, 2003b). Finally, Part 5 is a standard assumption (see, for instance, Section 5 of Bhandari & Russo (2019)), which guarantees that the policy parameterization is expressive enough to represent best responses.

**Assumption 4** (Convex Parameterization). *Given an inverse Markov game $\mathcal{M}^{-1}$, assume that for all players $i \in [n]$, states $\boldsymbol{s} \in \mathcal{S}$, and action profiles $\boldsymbol{a}, \boldsymbol{b} \in \mathcal{A}$, the per-state regret $\boldsymbol{\theta} \mapsto r_i(\boldsymbol{s}, \boldsymbol{b}_i, \boldsymbol{a}_{-i}; \boldsymbol{\theta}) - r_i(\boldsymbol{s}, \boldsymbol{a}; \boldsymbol{\theta})$ is convex.*

With these assumptions in hand, we face a convex gradient-dominated optimization problem, i.e., $\boldsymbol{\theta} \mapsto f(\boldsymbol{\theta}, \boldsymbol{x})$ is convex, for all $\boldsymbol{x} \in \mathcal{X}$, and $\boldsymbol{x} \mapsto f(\boldsymbol{\theta}, \boldsymbol{x})$ gradient-dominated, for all $\boldsymbol{\theta} \in \Theta$. As for normal-form games (see Remark 1), the set of inverse NE in Markov games is convex under Assumptions 3 and 4. Consequently, we can obtain polynomial-time convergence of stochastic gradient descent ascent (Algorithm 2) by slightly modifying known results (Daskalakis et al., 2020).

---

**Algorithm 2** Adversarial Inverse MARL

**Inputs:** $\Theta, \mathcal{P}, f_{\boldsymbol{\theta}}, f_{\boldsymbol{x}}, \eta_{\boldsymbol{\theta}}, \eta_{\boldsymbol{x}}, T, \boldsymbol{\theta}^{(0)}, \boldsymbol{x}^{(0)}, \boldsymbol{\pi}^{\dagger}$

**Outputs:** $(\boldsymbol{\theta}^{(t)}, \boldsymbol{x}^{(t)})_{t=0}^{T}$

1: **for** $t = 0, \ldots, T-1$ **do**
2: $\quad \boldsymbol{H} \sim \bigtimes_{i \in [n]} \nu^{(\boldsymbol{\pi}_i^{\boldsymbol{x}^{(t)}}, \boldsymbol{\pi}_{-i}^{\dagger})}, \boldsymbol{h}^{\dagger} \sim \nu^{\boldsymbol{\pi}^{\dagger}}$
3: $\quad \boldsymbol{\theta}^{(t+1)} \leftarrow \Pi_{\Theta} \left[ \boldsymbol{\theta}^{(t)} - \eta_{\boldsymbol{\theta}}^{(t)} \nabla_{\boldsymbol{\theta}} f_{\boldsymbol{\theta}}(\boldsymbol{\theta}^{(t)}, \boldsymbol{x}^{(t)}; \boldsymbol{H}, \boldsymbol{h}^{\dagger}) \right]$
4: $\quad \boldsymbol{x}^{(t+1)} \leftarrow \Pi_{\mathcal{P}} \left[ \boldsymbol{x}^{(t)} + \eta_{\boldsymbol{x}}^{(t)} \nabla_{\boldsymbol{x}} f_{\boldsymbol{x}}(\boldsymbol{\theta}^{(t)}, \boldsymbol{x}^{(t)}; \boldsymbol{H}, \boldsymbol{h}^{\dagger}) \right]$
5: **return** $(\boldsymbol{\theta}^{(t)}, \boldsymbol{x}^{(t)})_{t=0}^{T}$

---

Algorithm 2 requires an estimate of $\nabla f$ w.r.t. both $\boldsymbol{\theta}$ and $\boldsymbol{x}$. Under Part 3 of Assumption 3, the gradient of $f$ w.r.t. $\boldsymbol{x}$ can be obtained by the deterministic policy gradient theorem (Silver et al., 2014), while the gradient of $f$ w.r.t. $\boldsymbol{\theta}$ can be obtained by the linearity of the gradient and expectation operators. However, both of these gradients involve an expectation—over $H \sim \nu^{(\boldsymbol{\pi}_i^{\boldsymbol{x}}, \boldsymbol{\pi}_{-i}^{\dagger})}$ and $H^{\dagger} \sim \nu^{\boldsymbol{\pi}^{\dagger}}$. As such, we estimate them using simulated trajectories from the deviation history distribution $\boldsymbol{H} \doteq (\boldsymbol{h}^1, \ldots, \boldsymbol{h}^n)^T \sim \bigtimes_{i \in [n]} \nu^{(\boldsymbol{\pi}_i^{\boldsymbol{x}}, \boldsymbol{\pi}_{-i}^{\dagger})}$ and the equilibrium history distribution $\boldsymbol{h}^{\dagger} \sim \nu^{\boldsymbol{\pi}^{\dagger}}$, respectively. For a given such pair $(\boldsymbol{H}, \boldsymbol{h}^{\dagger})$, the cumulative regret gradient estimators $f_{\boldsymbol{\theta}}$ and $f_{\boldsymbol{x}}$ correspond to the gradients of the cumulative regrets between each deviation history $\boldsymbol{h}^i$ in $\boldsymbol{H}$ and $\boldsymbol{h}^{\dagger}$, and can be computed directly using the chain rule for derivatives, as we assume access to a differentiable game simulator.[8]

Finally, we define the *equilibrium distribution mismatch coefficient* $\|\partial \delta_{\mu}^{\boldsymbol{\pi}^{\dagger}}/\partial \mu\|_{\infty}$ as the Radon-Nikodym derivative of the state occupancy distribution of the NE $\boldsymbol{\pi}^{\dagger}$ w.r.t. the initial state distribution $\mu$. This coefficient, which measures the inherent difficulty of reaching states under $\boldsymbol{\pi}^{\dagger}$, is closely related to other distribution mismatch coefficients introduced in the analysis of policy gradient methods (Agarwal et al., 2020). With this definition in hand, we can finally show polynomial-time convergence of stochastic GDA (Algorithm 2) under Assumptions 3–4.

**Theorem 4.1.** *Under Assumptions 3–4, for all $\varepsilon \in (0, 1)$, if Algorithm 2 is run with inputs that satisfy $T \in \Omega\left(\varepsilon^{-10}\|\partial \delta_{\mu}^{\boldsymbol{\pi}^{\dagger}}/\partial \mu\|_{\infty}\right)$ and for all $t \in [T]$, $\eta_{\boldsymbol{y}}^{(t)} \asymp \varepsilon^4$ and $\eta_{\boldsymbol{\theta}}^{(t)} \asymp \varepsilon^8$, then the time-average of all parameters $\overline{\boldsymbol{\theta}^{(T)}} \doteq \frac{1}{T+1} \sum_{t=0}^{T} \boldsymbol{\theta}^{(t)}$ is an $\varepsilon$-inverse NE.*

## 5 SIMULACRAL LEARNING

In this section, we consider the more realistic setting in which we do not observe an equilibrium, but observe only sample histories $\{\boldsymbol{h}^{(k)}\}_k = \{(\boldsymbol{s}^{(t,k)}, \boldsymbol{a}^{(t,k)})_t\}_k \sim \nu^{\boldsymbol{\pi}^{\dagger}}$ associated with an *unobserved* equilibrium $\boldsymbol{\pi}^{\dagger}$. The problem of interest then becomes one of not only inferring parameter values from observed behavior, but of additionally finding equilibrium policies that generate the observed behavior, a solution which we refer to as a first-order simulacrum. A first-order simulacrum can be seen as a generalization of an inverse equilibrium, as it not only comprises parameters that rationalize

---

[8]For completeness, we show how to compute $f_{\boldsymbol{x}}$ and $f_{\boldsymbol{\theta}}$ in Section 7.5. In our experiments, however, as has become common practice in the literature (Mora et al., 2021), we compute these gradients by simply autodifferentiating the cumulative regret of any history w.r.t. the policy parameters using a library like Jax (Bradbury et al., 2018). We also show that under Assumption 3, $(f_{\boldsymbol{\theta}}, f_{\boldsymbol{x}})$ is an unbiased estimate of $(\nabla_{\boldsymbol{\theta}} f, \nabla_{\boldsymbol{x}} f)$ whose variance is bounded.

the observed histories, but also policies that mimic them in expectation. First-order simulacral learning is also known as *multiagent apprenticeship learning* (Abbeel & Ng, 2004; Yang et al., 2020).

---

**Algorithm 3** Adversarial Simulacral Learning

**Inputs:** $\Theta, \mathcal{P}, (\widehat{g_{\boldsymbol{\theta}}}, \widehat{g_{\boldsymbol{x}}}, \widehat{g_{\boldsymbol{y}}}), \eta_{\boldsymbol{\theta}}, \eta_{\boldsymbol{x}}, \eta_{\boldsymbol{y}}, T, \boldsymbol{\theta}^{(0)}, \boldsymbol{x}^{(0)}, \boldsymbol{y}^{(0)}, \{\boldsymbol{o}^{\dagger(k)}\}$

**Outputs:** $(\boldsymbol{\theta}^{(t)}, \boldsymbol{x}^{(t)}, \boldsymbol{y}^{(t)})_{t=0}^{T}$

1: **for** $t = 0, \ldots, T-1$ **do**

2: $\quad \boldsymbol{H} \sim \bigtimes_{i \in [n]} \nu^{(\boldsymbol{\pi}_i^{\boldsymbol{x}^{(t)}}, \boldsymbol{\pi}_{-i}^{\boldsymbol{y}^{(t)}})}, \boldsymbol{h} \sim \nu^{(\boldsymbol{\pi}^{\boldsymbol{x}^{(t)}})}$

3: $\quad \boldsymbol{\theta}^{(t+1)} \leftarrow \Pi_{\Theta} \left[ \boldsymbol{\theta}^{(t)} - \eta_{\boldsymbol{\theta}}^{(t)} \nabla_{\boldsymbol{\theta}} \widehat{g_{\boldsymbol{\theta}}}(\boldsymbol{\theta}^{(t)}, \boldsymbol{x}^{(t)}, \boldsymbol{y}^{(t)}; \boldsymbol{H}, \boldsymbol{h}) \right]$

4: $\quad \boldsymbol{x}^{(t+1)} \leftarrow \Pi_{\mathcal{P}} \left[ \boldsymbol{x}^{(t)} - \eta_{\boldsymbol{x}}^{(t)} \nabla_{\boldsymbol{x}} \widehat{g_{\boldsymbol{x}}}(\boldsymbol{\theta}^{(t)}, \boldsymbol{x}^{(t)}, \boldsymbol{y}^{(t)}; \boldsymbol{H}, \boldsymbol{h}) \right]$

5: $\quad \boldsymbol{y}^{(t+1)} \leftarrow \Pi_{\mathcal{P}} \left[ \boldsymbol{y}^{(t)} + \eta_{\boldsymbol{y}}^{(t)} \nabla_{\boldsymbol{y}} \widehat{g_{\boldsymbol{y}}}(\boldsymbol{\theta}^{(t)}, \boldsymbol{x}^{(t)}, \boldsymbol{y}^{(t)}; \boldsymbol{H}, \boldsymbol{h}) \right]$

6: **return** $(\boldsymbol{\theta}^{(t)}, \boldsymbol{x}^{(t)}, \boldsymbol{y}^{(t)})_{t=0}^{T}$

---

Even more generally, we might not have access to samples $\{\boldsymbol{h}^{\dagger(k)}\}_{k \in [\kappa]} \sim \nu^{\boldsymbol{\pi}^{\dagger}}$ from an equilibrium history distribution, but rather a lossy function of those histories according to some function $\rho : \mathcal{H} \rightarrow \mathcal{O}$ that produces *observations* $\{\boldsymbol{o}^{\dagger(k)}\}_{k \in [\kappa]} \doteq \{\rho(\boldsymbol{h}^{\dagger(k)})\}_{k \in [\kappa]} \sim \Xi^{\boldsymbol{\pi}^{\dagger}}$, distributed according to some *(pushforward) observation distribution* $\Xi^{\boldsymbol{\pi}} \in \Delta(\mathcal{O})$, parameterized by policy profile $\boldsymbol{\pi} \in \mathcal{P}$, where $\mathcal{O}$ is the observation space. This more general framework is very useful in applications where there are limitations on the data collection process: e.g., if there are game states at which some of the players' actions are unobservable, or when only an unfaithful function of them is available. Here, we seek to learn the more general notion of a *second-order simulacrum*.

Formally, an *inverse simulation* $\mathcal{I}^{-1} \doteq (\mathcal{M}^{\boldsymbol{\theta}^{\dagger}} \setminus \boldsymbol{\theta}^{\dagger}, \mathcal{O}, \Xi, \Xi^{\boldsymbol{\pi}^{\dagger}})$ is a tuple consisting of a Markov game form $\mathcal{M}^{\boldsymbol{\theta}^{\dagger}} \setminus \boldsymbol{\theta}^{\dagger}$ with unknown parameters $\boldsymbol{\theta}^{\dagger}$, an observation distribution $\Xi : \mathcal{P} \rightarrow \Delta(\mathcal{O})$ mapping policies to distributions over the *observation space* $\mathcal{O}$, and an observation distribution $\Xi^{\boldsymbol{\pi}^{\dagger}}$ for the *unobserved* behavioral policy $\boldsymbol{\pi}^{\dagger}$, which we assume is a Nash equilibrium. Our goal is to find an $(\varepsilon, \delta)$-*Nash simulacrum*, meaning a tuple of parameters and policies $(\boldsymbol{\theta}^*, \boldsymbol{\pi}^*) \in \Theta \times \mathcal{P}$ that $(\varepsilon, \delta)$-*simulates* the observations as a Nash equilibrium: i.e., $u_i(\boldsymbol{\pi}^*; \boldsymbol{\theta}^*) \geq \max_{\boldsymbol{\pi}_i \in \mathcal{P}_i} u_i(\boldsymbol{\pi}_i, \boldsymbol{\pi}_{-i}^*; \boldsymbol{\theta}^*) - \varepsilon$ and $\mathbb{E}_{(\boldsymbol{o}, \boldsymbol{o}^{\dagger}) \sim \Xi^{\boldsymbol{\pi}^*} \times \Xi^{\boldsymbol{\pi}^{\dagger}}} \left[ \|\boldsymbol{o} - \boldsymbol{o}^{\dagger}\|^2 \right] \leq \delta$. Theorem 5.1, which is analogous to Corollary 1, characterizes the set of Nash simulacra of an inverse simulation.

**Theorem 5.1.** *Given an inverse simulation* $\mathcal{I}^{-1}$, *for any* $\alpha, \beta > 0$, *the set of Nash simulacra of* $\mathcal{M}^{-1}$ *is equal to the set of minimizers of the following stochastic min-max optimization problem:*

$$\min_{\substack{\boldsymbol{\theta} \in \Theta \\ \boldsymbol{\pi} \in \mathcal{P}}} \varphi(\boldsymbol{\theta}, \boldsymbol{\pi}) = \min_{\substack{\boldsymbol{\theta} \in \Theta \\ \boldsymbol{\pi} \in \mathcal{P}}} \max_{\boldsymbol{\rho} \in \mathcal{P}} g(\boldsymbol{\theta}, \boldsymbol{\pi}, \boldsymbol{\rho}) \doteq \alpha \mathop{\mathbb{E}}_{(\boldsymbol{o}, \boldsymbol{o}^{\dagger}) \sim \Xi^{\boldsymbol{\pi}} \times \Xi^{\boldsymbol{\pi}^{\dagger}}} \left[ \|\boldsymbol{o} - \boldsymbol{o}^{\dagger}\|^2 \right] + \beta \psi(\boldsymbol{\pi}, \boldsymbol{\rho}; \boldsymbol{\theta}) \quad (3)$$

To tackle simulacral learning, we approximate $g$ via realized observation samples $\{\boldsymbol{o}^{\dagger(k)}\} \sim \Xi^{\boldsymbol{\pi}^{\dagger}}$, based on which we compute the empirical learning loss $\widehat{g}(\boldsymbol{\theta}, \boldsymbol{\pi}, \boldsymbol{\rho}) \doteq \alpha \mathbb{E}_{\boldsymbol{o} \sim \Xi^{\boldsymbol{\pi}}} \left[ 1/\kappa \sum_{k=1}^{\kappa} \|\boldsymbol{o} - \boldsymbol{o}^{\dagger(k)}\|^2 \right] + \beta \psi(\boldsymbol{\pi}, \boldsymbol{\rho}; \boldsymbol{\theta})$. Additionally, as in the previous section, we once again restrict policies to lie within a parametric class of policies $\mathcal{P}^{\mathcal{X}}$, redefine $g(\boldsymbol{\theta}, \boldsymbol{x}, \boldsymbol{y}) \doteq g(\boldsymbol{\theta}, \boldsymbol{\pi}^{\boldsymbol{x}}, \boldsymbol{\pi}^{\boldsymbol{y}})$ and $\widehat{g}(\boldsymbol{\theta}, \boldsymbol{x}, \boldsymbol{y}) \doteq \widehat{g}(\boldsymbol{\theta}, \boldsymbol{\pi}^{\boldsymbol{x}}, \boldsymbol{\pi}^{\boldsymbol{y}})$, and solve the ensuing optimization problem over the empirical learning loss $\min_{(\boldsymbol{\theta}, \boldsymbol{x}) \in \Theta \times \mathcal{X}} \max_{\boldsymbol{y} \in \mathcal{X}} \widehat{g}(\boldsymbol{\theta}, \boldsymbol{x}, \boldsymbol{y})$.

In general, this stochastic min-max optimization is non-convex non-concave. By Assumption 3, however, the function $\boldsymbol{y} \mapsto g(\boldsymbol{\theta}, \boldsymbol{x}, \boldsymbol{y})$ is gradient dominated, for all $\boldsymbol{\theta} \in \Theta$ and $\boldsymbol{x} \in \mathcal{X}$. Nevertheless, it is not possible to guarantee that $(\boldsymbol{\theta}, \boldsymbol{x}) \mapsto g(\boldsymbol{\theta}, \boldsymbol{x}, \boldsymbol{y})$ is convex or gradient dominated, for all $\boldsymbol{y} \in \mathcal{Y}$, without overly restrictive assumptions. This claim is intuitive, since the computation of an inverse simulacrum involves computing a Nash equilibrium policy, which in general is a PPAD-complete problem (Daskalakis et al., 2009; Foster et al., 2023). Finally, defining gradient estimators as we did in Section 4, to obtain gradient estimators $(\widehat{g_{\boldsymbol{\theta}}}, \widehat{g_{\boldsymbol{x}}}, \widehat{g_{\boldsymbol{y}}})(\boldsymbol{\theta}, \boldsymbol{x}, \boldsymbol{y}; \boldsymbol{H}, \boldsymbol{h}^{\boldsymbol{x}})$ from samples histories $\boldsymbol{H} \sim \bigtimes_{i \in [n]} \nu^{(\boldsymbol{\pi}_i^{\boldsymbol{x}}, \boldsymbol{\pi}_{-i}^{\boldsymbol{y}})}$ and $\boldsymbol{h}^{\boldsymbol{x}} \sim \nu^{\boldsymbol{\pi}^{\boldsymbol{x}}}$, we can use Algorithm 3 to compute a local solution of Equation (3) from polynomially-many observations.

**Theorem 5.2.** *Suppose that Assumption 3 holds, and that for all* $\boldsymbol{\pi}^{\boldsymbol{x}} \in \mathcal{P}^{\mathcal{X}}$, $\Xi^{\boldsymbol{\pi}^{\boldsymbol{x}}}$ *is twice continuously differentiable in* $\boldsymbol{x}$. *For any* $\varepsilon \in (0, 1)$, *if Algorithm 3 is run with inputs that satisfy* $T \in \Omega \left( \sigma^2/\varepsilon^{10} \|\partial \delta_{\mu}^{\boldsymbol{\pi}^*}/\partial \mu\|_{\infty} \right)$ *and for all* $t \in [T]$, $\eta_{\boldsymbol{y}}^{(t)} \asymp \varepsilon^4$ *and* $\eta_{\boldsymbol{\theta}}^{(t)} \asymp \varepsilon^8$, *then the best iterate*

$(\boldsymbol{\theta}_{\text{best}}, \boldsymbol{x}_{\text{best}})$ *converges to an $\varepsilon$-stationary point of $\varphi$ (defined in Section 7.2). Additionally, for any $\zeta, \xi \geq 0$, it holds with probability $1 - \zeta$ that $\widehat{\varphi}(\boldsymbol{\theta}_{\text{best}}^{(T)}, \boldsymbol{x}_{\text{best}}^{(T)}) - \varphi(\boldsymbol{\theta}_{\text{best}}^{(T)}, \boldsymbol{x}_{\text{best}}^{(T)}) \leq \xi$ if the number of sample observations $\kappa \in \Omega(1/\xi^2 \log(1/\zeta))$.*

## 6 EXPERIMENTS

We run two sets of experiments with the aim of answering two questions. Our first goal is to understand the extent to which our algorithms are able to compute inverse Nash equilibria, if any, beyond our theoretical guarantees. Our second goal is to understand the ability of game-theoretic models to make predictions about the future.[9]

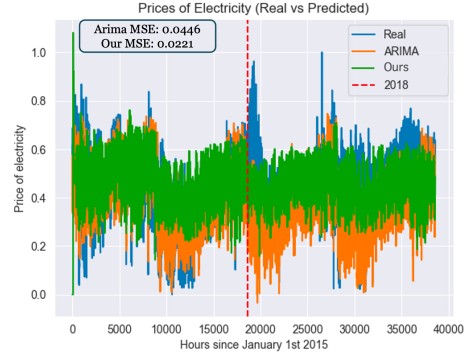

In our first set of experiments, we consider five types of economic games whose equilibria and payoffs have different properties. The first three are Fisher market (FM) games, which are zero-sum, between sellers and buyers engaged in trading goods. These games can be categorized based on the buyers' utility functions as linear, Cobb-Douglas, or Leontief (Cheung et al., 2013). We then consider two general-sum economic games, which model competition between two firms, namely Cournot competition and Bertrand oligopoly. When budgets are the only parameters we seek to recover, our min-max formulation is convex-concave, because the players' payoffs are concave in their actions, and affine in their budgets, and hence the regret of players is also affine in the players' budgets. In addition, in both the Cournot competition and Bertrand oligopoly games, regret is again convex in the parameters of the game. Finally, all the games we study are concave, with the exception of the Bertrand oligopoly game, and the equilibria are unique in the

Figure 1: Hourly prices in the Spanish electricity market from January 2015 to December 2020. The Nash simulacrum achieves a MSE that is twice as low as that of the ARIMA method.

Cobb-Douglas FM, Cournot competition, and Bertrand oligopoly games. In each experiment, we generate 500 synthetic game instances, for which the true parameters are known, and use Algorithm 1 (which does not rely on this knowledge) to compute an inverse NE for each. We record whether our algorithm recovers the true parameters of the market and whether it finds an inverse NE (i.e., average exploitability). We summarize our findings for the FM games in Table 2. We find that our algorithm recovers the true parameters more often when budgets are the only parameters we seek to recover, as opposed to both budgets and types; but even in non-convex-concave case, our algorithm is still able to approximate inverse NE over 80% of the time. In settings where the equilibria are unique, we recover true parameters most often, while the worst performance is on Leontief FM games, where payoffs are not differentiable.

| Game Parameters | Budgets | | | Types + Budgets | | |
|---|---|---|---|---|---|---|
| Fisher Market Type | Linear | Leontief | CD | Linear | Leontief | CD |
| % Parameters Recovered | 100% | 36.8% | 100% | 12% | 1% | 99.6% |
| Average Exploitability | 0.0018 | 0.2240 | 0.0004 | 0.0119 | 0.1949 | 0.0004 |

| | Cournot | Bertrand |
|---|---|---|
| % Parameters Recovered | 95.2% | 78% |
| Average Exploitability | 0.0000 | 0.0011 |

Table 2: The percentage of games for which we recovered the true parameters and the average exploitabilities of the observed equilibrium evaluated w.r.t the computed inverse Nash equilibrium. In our second set of experiments, we model the Spanish electricity market as a stochastic Fisher market game between electricity re-sellers and consumers. In this game, the state comprises the supply of each good and the consumers' budgets, while the re-sellers' actions are to set prices in today's spot market and tomorrow's day ahead market, and the consumers' actions are their electricity demands. We assume the consumers utilities are linear; this choice is suited to modeling the substitution effect between electricity today and electricity tomorrow. Using publicly available hourly Spanish electricity prices and aggregate demand data from Kaggle, we compute a simulacrum of the game that seeks to replicate these observations from January 2015 to December 2016. We also train an ARIMA model on the same data, and run a hyperparameter search for both algorithms using data from January 2017 to December 2018. After picking hyperparameters, we then retrain both models on the data between January 2015 to December 2018, and predict prices up to December 2018. We also compute the mean squared error (MSE) of both methods using January 2018 to December 2020 as a test set. We show the predictions of both methods in Figure 1. To summarize, we find that the simulacrum makes predictions whose MSE is twice as low.

---

[9]Our code can be found here.

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
