# 7 APPENDIX

## 7.1 ADDITIONAL PRELIMINARY DEFINITIONS

**Notation** We use caligraphic uppercase letters to denote sets (e.g., $\mathcal{X}$), bold uppercase letters to denote matrices (e.g., $\boldsymbol{X}$), bold lowercase letters to denote vectors (e.g., $\boldsymbol{p}$), lowercase letters to denote scalar quantities (e.g., $x$), and uppercase letters to denote random variables (e.g., $X$). We denote the $i$th row vector of a matrix (e.g., $\boldsymbol{X}$) by the corresponding bold lowercase letter with subscript $i$ (e.g., $\boldsymbol{x}_i$) and the $j$th entry of a vector (e.g., $\boldsymbol{p}$ or $\boldsymbol{x}_i$) by the corresponding Roman lowercase letter with subscript $j$ (e.g., $p_j$ or $x_{ij}$). We denote functions by a letter determined by the value of the function: e.g., $f$ if the mapping is scalar-valued, $\boldsymbol{f}$ if the mapping is vector-valued, and $\mathcal{F}$ if the mapping is set-valued. We denote the set of natural numbers by $\mathbb{N}$ and the set of real numbers by $\mathbb{R}$. We denote the positive and strictly positive elements of a set by a $+$ and $++$ subscript, respectively, e.g., $\mathbb{R}_+$ and $\mathbb{R}_{++}$. For any set $\mathcal{C}$, we denote its diameter $\max_{\boldsymbol{c}, \boldsymbol{c}' \in \mathcal{C}} \|\boldsymbol{c} - \boldsymbol{c}'\|$ by $\mathrm{diam}(\mathcal{C})$.

## 7.2 OMMITED PROOFS

**Theorem 3.1.** *The set of inverse NE of $\mathcal{G}^{-1}$ is the set of parameter profiles $\boldsymbol{\theta} \in \Theta$ that solve the optimization problem $\min_{\boldsymbol{\theta} \in \Theta} \varphi(\boldsymbol{x}^\dagger; \boldsymbol{\theta})$, or equivalently, this min-max optimization problem:*

$$\min_{\boldsymbol{\theta} \in \Theta} \max_{\boldsymbol{y} \in \mathcal{X}} f(\boldsymbol{\theta}, \boldsymbol{y}) \doteq \psi(\boldsymbol{x}^\dagger, \boldsymbol{y}; \boldsymbol{\theta}) = \sum_{i \in [n]} \left[ u_i(\boldsymbol{y}_i, \boldsymbol{x}^\dagger_{-i}; \boldsymbol{\theta}) - u_i(\boldsymbol{x}^\dagger; \boldsymbol{\theta}) \right] \tag{1}$$

*Proof of Theorem 3.1.* Notice that for all action profiles $\boldsymbol{x} \in \mathcal{X}$ and parameter profile $\boldsymbol{\theta} \in \Theta$, we have:

$$\max_{\boldsymbol{y} \in \mathcal{X}} f(\boldsymbol{\theta}, \boldsymbol{y}) = \max_{\boldsymbol{y} \in \mathcal{X}} \sum_{i \in [n]} \left[ u_i(\boldsymbol{y}_i, \boldsymbol{x}^\dagger; \boldsymbol{\theta}) - u_i(\boldsymbol{x}^\dagger; \boldsymbol{\theta}) \right] \tag{4}$$

$$= \sum_{i \in [n]} \left[ \max_{\boldsymbol{y}_i \in \mathcal{X}_i} u_i(\boldsymbol{y}_i, \boldsymbol{x}^\dagger_{-i}; \boldsymbol{\theta}) - u_i(\boldsymbol{x}^\dagger; \boldsymbol{\theta}) \right] \tag{5}$$

$$\geq 0 \tag{6}$$

Additionally, note that under our assumption the set of inverse-NE is non-empty. Hence, there exists a parameter profile $\widehat{\boldsymbol{\theta}} \in \Theta$ such that $\widehat{\boldsymbol{\theta}}$ is an inverse-NE, of $\mathcal{G}^{-1}$, i.e., for all players $i \in i$:

$$\max_{\boldsymbol{y}_i \in \mathcal{X}_i} u_i(\boldsymbol{y}_i, \boldsymbol{x}^\dagger_{-i}; \widehat{\boldsymbol{\theta}}) - u_i(\boldsymbol{x}^\dagger; \widehat{\boldsymbol{\theta}}) = 0 \tag{7}$$

Summing the above equality across all players, we then have:

$$\sum_{i \in [n]} \max_{\boldsymbol{y}_i \in \mathcal{X}_i} u_i(\boldsymbol{y}_i, \boldsymbol{x}^\dagger_{-i}; \widehat{\boldsymbol{\theta}}) - u_i(\boldsymbol{x}^\dagger; \widehat{\boldsymbol{\theta}}) = 0 \tag{8}$$

This means that the minimum of $\max_{\boldsymbol{y} \in \mathcal{X}} f(\boldsymbol{\theta}, \boldsymbol{y})$ is achieved at 0, since for all $\boldsymbol{\theta} \in \Theta$, $\max_{\boldsymbol{y} \in \mathcal{X}} f(\boldsymbol{\theta}, \boldsymbol{y}) \geq 0$.

Let $(\boldsymbol{\theta}^*, \boldsymbol{x}^*)$ be any optimal solution to $\min_{\boldsymbol{\theta} \in \Theta} \max_{\boldsymbol{y} \in \mathcal{X}} f(\boldsymbol{\theta}, \boldsymbol{y})$. We will show that $\boldsymbol{\theta}^*$ is an inverse NE of $\mathcal{G}^{-1}$.

Since the minimum of $\max_{\boldsymbol{y} \in \mathcal{X}} f(\boldsymbol{\theta}, \boldsymbol{y})$ is achieved at 0, it follows that

$$\max_{\boldsymbol{y} \in \mathcal{X}} f(\boldsymbol{\theta}^*, \boldsymbol{y}) = \sum_{i \in [n]} \underbrace{\max_{\boldsymbol{y}_i \in \mathcal{X}_i} u_i(\boldsymbol{y}_i, \boldsymbol{x}^\dagger_{-i}; \boldsymbol{\theta}^*) - u_i(\boldsymbol{x}^\dagger; \boldsymbol{\theta}^*)}_{\geq 0} = 0 \tag{9}$$

But then, since for all players $i \in [n]$, $\max_{\boldsymbol{y}_i \in \mathcal{X}_i} u_i(\boldsymbol{y}_i, \boldsymbol{x}^\dagger_{-i}; \boldsymbol{\theta}^*) - u_i(\boldsymbol{x}^\dagger; \boldsymbol{\theta}^*) \geq 0$, it must hold that:

$$\max_{\boldsymbol{y}_i \in \mathcal{X}_i} u_i(\boldsymbol{y}_i, \boldsymbol{x}^\dagger_{-i}; \boldsymbol{\theta}^*) - u_i(\boldsymbol{x}^\dagger; \boldsymbol{\theta}^*) = 0 \tag{10}$$

$$\max_{\boldsymbol{y}_i \in \mathcal{X}_i} u_i(\boldsymbol{y}_i, \boldsymbol{x}^\dagger_{-i}; \boldsymbol{\theta}^*) = u_i(\boldsymbol{x}^\dagger; \boldsymbol{\theta}^*) \tag{11}$$

hence proving that $\boldsymbol{x}^\dagger$ is a Nash equilibrium under parameters $\boldsymbol{\theta}^*$, i.e., $\boldsymbol{\theta}^*$ is an inverse Nash equilibrium. $\square$

**Theorem 3.2** (Detailed Statement). *Suppose that Assumptions 1–2 hold. If Algorithm 1 is run with inputs that satisfy for all $\varepsilon \geq 0$, $t \in [T]$ $\eta_{\boldsymbol{y}}^{(t)} = \eta_{\boldsymbol{\theta}}^{(t)} = \frac{2\sum_{i\in[n]} \ell_{\nabla u_i}}{t}$, and $T \geq \frac{\mathrm{diam}(\Theta \times \mathcal{X})}{\varepsilon^2}$ for $\varepsilon \geq 0$, then the time-average of all parameters $\overline{\boldsymbol{\theta}^{(T)}} \doteq \frac{1}{T+1}\sum_{t=0}^{T} \boldsymbol{\theta}^{(t)}$ is an $\varepsilon$-inverse NE, i.e., $\psi(\boldsymbol{x}^\dagger, \boldsymbol{y}; \overline{\boldsymbol{\theta}^{(T)}}) - \min_{\boldsymbol{\theta}\in\Theta}\max_{\boldsymbol{y}\in\mathcal{X}}\psi(\boldsymbol{x}^\dagger, \boldsymbol{y}; \boldsymbol{\theta}) \leq \varepsilon$.*

*Proof of Theorem 3.2.* The theorem is a direct consequence of Result 3.1 of Nemirovski et al. (2009).
□

**Theorem 4.1** (Detailed Statement). *Under Assumption 3, if Algorithm 2 is run with inputs that satisfy for all $t \in T$, $\varepsilon \in (0,1)$, $\eta_{\boldsymbol{y}}^{(t)} \asymp \frac{\epsilon^4\left(\left\|\partial\delta_\mu^{\boldsymbol{\pi}^*}/\partial\mu\right\|_\infty/(1-\gamma)\cdot\mu\right)^2}{\ell_{\nabla f}^3\left(\ell_{\nabla f}^2+\sigma^2\right)\left(\ell_f/\ell_{\nabla f}+1\right)}$, $\eta_{\boldsymbol{\theta}}^{(t)} \asymp \frac{\varepsilon^8\left(\left\|\partial\delta_\mu^{\boldsymbol{\pi}^*}/\partial\mu\right\|_\infty/(1-\gamma)\cdot\mu\right)^4}{\ell_{\nabla f}^5\ell_f\left(\frac{\ell_f}{\ell_{\nabla f}}^2+1\right)^4\left(\ell_f^2+\sigma^2\right)^{3/2}} \wedge \frac{\varepsilon^2}{\ell_{\nabla f}\left(\ell_f^2+\sigma^2\right)}$, and $T \geq \left(1+\frac{\ell_f}{2\ell_{\nabla f}}\right)^{-1}\frac{\ell_f\mathrm{diam}(\mathcal{X}\times\Theta)}{\varepsilon^2\eta_{\boldsymbol{\theta}}^{(t)}}$, then the time-average of all parameters $\overline{\boldsymbol{\theta}^{(T)}} \doteq \frac{1}{T+1}\sum_{t=0}^{T}\boldsymbol{\theta}^{(t)}$ is a $\varepsilon$-inverse NE, i.e., $\max_{\boldsymbol{x}\in\mathcal{X}}\psi(\boldsymbol{x}^\dagger, \boldsymbol{x}; \overline{\boldsymbol{\theta}^{(T)}}) - \min_{\boldsymbol{\theta}\in\Theta}\max_{\boldsymbol{x}\in\mathcal{X}}\psi(\boldsymbol{x}^\dagger, \boldsymbol{x}; \boldsymbol{\theta}) \leq \varepsilon$.*

*Proof of Theorem 4.1.* Firstly, note that under Assumption 4, $f(\boldsymbol{\theta}, \boldsymbol{x})$ is 1-gradient-dominated in $\boldsymbol{\theta}$ since it is convex in $\boldsymbol{\theta}$ for all $\boldsymbol{y} \in \mathcal{X}$ (see Definition 2 of Bhandari & Russo (2019)).

Now, define the *equilibrium distribution mismatch coefficient* $\|\partial\delta_\mu^{\boldsymbol{\pi}^\dagger}/\partial\mu\|_\infty$ as the Radon-Nikodym derivative of the state-visitation distribution of the Nash equilibrium $\boldsymbol{\pi}^\dagger$ w.r.t. the initial state distribution $\mu$. Under Assumption 3, by Theorem 2 and Theorem 4 of Bhandari & Russo (2019), we also have that $f(\boldsymbol{\theta}, \boldsymbol{x})$ is $\left(\|\partial\delta_\mu^{\boldsymbol{\pi}^\dagger}/\partial\mu\|_\infty/(1-\gamma)\cdot\mu\right)$-gradient-dominated in $\boldsymbol{x}$ for all $\boldsymbol{\theta} \in \Theta$.

Moreover, according to analysis in Section 7.5, the variance of the gradient estimator is bounded. Hence, under our Theorem's assumptions, the assumptions of Theorem 2 of Daskalakis et al. (2020) are satisfied, and we have:

$$\frac{1}{T+1}\sum_{t=0}^{T}\max_{\boldsymbol{x}\in\mathcal{X}}\psi(\boldsymbol{x}^\dagger, \boldsymbol{x}; \boldsymbol{\theta}^{(t)}) - \min_{\boldsymbol{\theta}\in\Theta}\max_{\boldsymbol{x}\in\mathcal{X}}\psi(\boldsymbol{x}^\dagger, \boldsymbol{x}; \boldsymbol{\theta}) \leq \varepsilon \quad (12)$$

Note that since $\psi(\boldsymbol{x}^\dagger, \boldsymbol{x}, \boldsymbol{\theta})$ is convex in $\boldsymbol{\theta}$ for all $\boldsymbol{x} \in \mathcal{X}$, then $\boldsymbol{x} \mapsto \max_{\boldsymbol{x}\in\mathcal{X}}\psi(\boldsymbol{x}^\dagger, \boldsymbol{x}; \boldsymbol{\theta}^{(t)})$ is convex by Dankin's theorem Danskin (1966). Hence, using convexity, we obtain the theorem's result:

$$\max_{\boldsymbol{x}\in\mathcal{X}}\psi(\boldsymbol{x}^\dagger, \boldsymbol{x}; \frac{1}{T+1}\sum_{t=0}^{T}\boldsymbol{\theta}^{(t)}) - \min_{\boldsymbol{\theta}\in\Theta}\max_{\boldsymbol{x}\in\mathcal{X}}\psi(\boldsymbol{x}^\dagger, \boldsymbol{x}; \boldsymbol{\theta}) \leq \varepsilon \quad (13)$$

□

Theorem 4.1 tells us that in inverse Markov games satisfying Assumption 3, an $\varepsilon$-inverse NE can be computed in $T \asymp \sigma^2/\varepsilon^{10}\|\partial\delta_\mu^{\boldsymbol{\pi}^*}/\partial\mu\|_\infty$. One way to interpret this result is that the closer the initial state distribution is to the equilibrium state visitation distribution, i.e., the smaller $\left\|\partial\delta_\mu^{\boldsymbol{\pi}^*}/\partial\mu\right\|$ is, and the smaller the variance $\sigma^2$ of the gradient estimators is, the faster the convergence. On the other hand, if any state that is visited with strictly positive probability by the Nash equilibrium policy is not part of the support of the initial state distribution, then $\left\|\partial\delta_\mu^{\boldsymbol{\pi}^*}/\partial\mu\right\| \to \infty$, and the convergence bound degrades arbitrarily.

**Theorem 5.1.** *Given an inverse simulation $\mathcal{I}^{-1}$, for any $\alpha, \beta > 0$, the set of Nash simulacra of $\mathcal{M}^{-1}$ is equal to the set of minimizers of the following stochastic min-max optimization problem:*

$$\min_{\substack{\boldsymbol{\theta}\in\Theta \\ \boldsymbol{\pi}\in\mathcal{P}}}\varphi(\boldsymbol{\theta}, \boldsymbol{\pi}) = \min_{\substack{\boldsymbol{\theta}\in\Theta \\ \boldsymbol{\pi}\in\mathcal{P}}}\max_{\boldsymbol{\rho}\in\mathcal{P}} g(\boldsymbol{\theta}, \boldsymbol{\pi}, \boldsymbol{\rho}) \doteq \alpha \mathop{\mathbb{E}}_{(\boldsymbol{o},\boldsymbol{o}^\dagger)\sim\Xi^{\boldsymbol{\pi}}\times\Xi^{\boldsymbol{\pi}^\dagger}}\left[\left\|\boldsymbol{o}-\boldsymbol{o}^\dagger\right\|^2\right] + \beta\psi(\boldsymbol{\pi}, \boldsymbol{\rho}; \boldsymbol{\theta}) \quad (3)$$

*Proof of Theorem 5.1.* Fix $\alpha, \beta > 0$. Let $(\boldsymbol{\theta}^*, \boldsymbol{\pi}^*)$ be the optimal solutions to the above optimization problem. Notice that for all policy profiles $\boldsymbol{\pi} \in \mathcal{P}$ and parameter profiles $\boldsymbol{\theta} \in \Theta$, we have $\alpha \left\| \boldsymbol{\pi} - \boldsymbol{\pi}^\dagger \right\|_2^2 \geq 0$ by the definition of the euclidean norm. Additionally, we have for all $\boldsymbol{\pi} \in \mathcal{P}$, $\boldsymbol{\theta} \in \Theta$:

$$\max_{\boldsymbol{\rho} \in \mathcal{P}} \beta \psi(\boldsymbol{\pi}, \boldsymbol{\rho}; \boldsymbol{\theta}) = \max_{\boldsymbol{\rho} \in \mathcal{P}} \beta \sum_{i \in [n]} [u_i(\boldsymbol{\rho}_i, \boldsymbol{\pi}; \boldsymbol{\theta}) - u_i(\boldsymbol{\pi}; \boldsymbol{\theta})] \tag{14}$$

$$= \beta \sum_{i \in [n]} \left[ \max_{\boldsymbol{\rho}_i \in \mathcal{P}_i} u_i(\boldsymbol{\rho}_i, \boldsymbol{\pi}_{-i}; \boldsymbol{\theta}) - u_i(\boldsymbol{\pi}; \boldsymbol{\theta}) \right] \tag{15}$$

$$\geq 0 \tag{16}$$

Hence, we have:

$$\max_{\boldsymbol{\rho} \in \mathcal{P}} g(\boldsymbol{\theta}, \boldsymbol{\pi}, \boldsymbol{\rho}) = \max_{\boldsymbol{\rho} \in \mathcal{P}} \left\{ \alpha \mathop{\mathbb{E}}_{(\boldsymbol{o}, \boldsymbol{o}^\dagger) \sim \Xi^{\boldsymbol{\pi}} \times \Xi^{\boldsymbol{\pi}^\dagger}} \left[ \left\| \boldsymbol{o} - \boldsymbol{o}^\dagger \right\|^2 \right] + \beta \psi(\boldsymbol{\pi}, \boldsymbol{\rho}; \boldsymbol{\theta}) \right\} \tag{17}$$

$$= \alpha \mathop{\mathbb{E}}_{(\boldsymbol{o}, \boldsymbol{o}^\dagger) \sim \Xi^{\boldsymbol{\pi}} \times \Xi^{\boldsymbol{\pi}^\dagger}} \left[ \left\| \boldsymbol{o} - \boldsymbol{o}^\dagger \right\|^2 \right] + \max_{\boldsymbol{\rho} \in \mathcal{P}} \beta \psi(\boldsymbol{\pi}, \boldsymbol{\rho}; \boldsymbol{\theta}) \tag{18}$$

$$\geq \alpha(0) + \beta(0) = 0 \tag{19}$$

Additionally, note that under our assumption the set of inverse-NE is non-empty. Hence, there exists a tuple of parameter and action profiles $(\boldsymbol{\theta}^*, \boldsymbol{\pi}^*)$ such that $\boldsymbol{\pi}^* = \boldsymbol{\pi}^\dagger$, and $\boldsymbol{\theta}^*$ is an inverse-NE, of $(n, m, \mathcal{P}, \Theta, \boldsymbol{u}, \boldsymbol{\pi}^\dagger)$:

$$\max_{\boldsymbol{\rho} \in \mathcal{P}} g(\boldsymbol{\theta}^*, \boldsymbol{\pi}^*, \boldsymbol{\rho}) \tag{20}$$

$$= \max_{\boldsymbol{\rho} \in \mathcal{P}} \left\{ \alpha \mathop{\mathbb{E}}_{(\boldsymbol{o}, \boldsymbol{o}^\dagger) \sim \Xi^{\boldsymbol{\pi}^*} \times \Xi^{\boldsymbol{\pi}^\dagger}} \left[ \left\| \boldsymbol{o} - \boldsymbol{o}^\dagger \right\|^2 \right] + \beta \psi(\boldsymbol{\pi}^\dagger, \boldsymbol{\rho}; \boldsymbol{\theta}^*) \right\} \tag{21}$$

$$= \alpha \mathop{\mathbb{E}}_{(\boldsymbol{o}, \boldsymbol{o}^\dagger) \sim \Xi^{\boldsymbol{\pi}^*} \times \Xi^{\boldsymbol{\pi}^\dagger}} \left[ \left\| \boldsymbol{o} - \boldsymbol{o}^\dagger \right\|^2 \right] + \beta \max_{\boldsymbol{\rho} \in \mathcal{P}} \psi(\boldsymbol{\pi}^\dagger, \boldsymbol{\rho}; \boldsymbol{\theta}^*) \tag{22}$$

$$= \alpha(0) + \beta(0) = 0 \tag{23}$$

where the final line follows from the definition of the inverse Nash equilibrium, i.e., $\psi(\boldsymbol{\pi}^\dagger, \boldsymbol{\rho}; \boldsymbol{\theta}^*) = 0$.

This in turn means that the minimum of $\max_{\boldsymbol{\rho} \in \mathcal{P}} g(\boldsymbol{\theta}, \boldsymbol{\pi}, \boldsymbol{\rho})$ is achieved at 0.

Finally, we show that any tuple $(\boldsymbol{\theta}^*, \boldsymbol{\pi}^*)$ of parameter and action profiles which are a minimum of $\max_{\boldsymbol{\rho} \in \mathcal{P}} g(\boldsymbol{\theta}, \boldsymbol{\pi}, \boldsymbol{\rho})$, i.e. $(\boldsymbol{\theta}^*, \boldsymbol{\pi}^*) \in \arg \min_{\substack{\boldsymbol{\theta} \in \Theta \\ \boldsymbol{\pi} \in \mathcal{P}}} \max_{\boldsymbol{\rho} \in \mathcal{P}} g(\boldsymbol{\theta}, \boldsymbol{\pi}, \boldsymbol{\rho})$ respectively correspond to a tuple $(\boldsymbol{\theta}^*, \boldsymbol{\pi}^*)$ such that $\mathbb{E}_{(\boldsymbol{o}, \boldsymbol{o}^\dagger) \sim \Xi^{\boldsymbol{\pi}^*} \times \Xi^{\boldsymbol{\pi}^\dagger}}$, and $\boldsymbol{\theta}^*$ is an inverse-NE of $(n, m, \mathcal{P}, \Theta, \boldsymbol{u}, \boldsymbol{\pi}^*)$.

Recall that, by Equation (23), we have:

$$\min_{\substack{\boldsymbol{\theta} \in \Theta \\ \boldsymbol{\pi} \in \mathcal{P}}} \max_{\boldsymbol{\rho} \in \mathcal{P}} g(\boldsymbol{\theta}, \boldsymbol{\pi}, \boldsymbol{\rho}) = \min_{\substack{\boldsymbol{\theta} \in \Theta \\ \boldsymbol{\pi} \in \mathcal{P}}} \max_{\boldsymbol{\rho} \in \mathcal{P}} \left\{ \alpha \left\| \boldsymbol{\pi}^\dagger - \boldsymbol{\pi} \right\|_2^2 + \beta \psi(\boldsymbol{\pi}, \boldsymbol{\rho}; \boldsymbol{\theta}) \right\} \tag{24}$$

$$= \min_{\substack{\boldsymbol{\theta} \in \Theta \\ \boldsymbol{\pi} \in \mathcal{P}}} \left\{ \alpha \underbrace{\left\| \boldsymbol{\pi}^\dagger - \boldsymbol{\pi} \right\|_2^2}_{\geq 0} + \underbrace{\max_{\boldsymbol{\rho} \in \mathcal{P}} \beta \psi(\boldsymbol{\pi}, \boldsymbol{\rho}; \boldsymbol{\theta})}_{\geq 0} \right\} \tag{25}$$

$$= 0 \tag{26}$$

Hence, it must be that $\left\| \boldsymbol{\pi}^\dagger - \boldsymbol{\pi}^* \right\|_2^2 = 0$ and $\max_{\boldsymbol{\rho} \in \mathcal{P}} \psi(\boldsymbol{\pi}^*, \boldsymbol{\rho}; \boldsymbol{\theta}^*) = 0$, proving the desired result. $\qquad \square$

As the computation of a simulacrum is in general a non-convex-non-concave problem, we cannot compute a solution to the min-max optimization in polynomial-time, however, we can obtain best

iterate convergence to a stationary point of the *Moreau envelope of the empirical exploitability* $\widetilde{\varphi}(\boldsymbol{\theta}, \boldsymbol{x}) \doteq \min_{(\boldsymbol{\theta}', \boldsymbol{x}') \in \Theta \times \mathcal{X}} \left\{ \varphi(\boldsymbol{\theta}, \boldsymbol{x}) + \ell_{\nabla \widehat{g}} \left\| (\boldsymbol{\theta}, \boldsymbol{x}) - (\boldsymbol{\theta}', \boldsymbol{x}') \right\|^2 \right\}$, i.e., a point $(\boldsymbol{\theta}, \boldsymbol{x}) \in \Theta \times \mathcal{X}$ s.t. $\|\nabla \widetilde{\varphi}(\boldsymbol{\theta}, \boldsymbol{x})\| = 0$ under suitable assumptions satisfied by a large class of Markov games including discrete state and action space Markov games.[10]

**Theorem 5.2** (Detailed Statement). *Suppose that Assumption 3 holds, and assume in addition that for all $\boldsymbol{\pi}^{\boldsymbol{x}} \in \mathcal{P}^{\mathcal{X}}$, $\Xi^{\boldsymbol{\pi}^{\boldsymbol{x}}}$ is twice continuously differentiable in $\boldsymbol{x}$. If Algorithm 3*

*is run with inputs that satisfy for all $t \in [T] \; \varepsilon \in (0,1)$, $\eta_{\boldsymbol{y}}^{(t)} \asymp \dfrac{\epsilon^4 \left( \left\| \partial \delta_\mu^{\boldsymbol{\pi}^*} / \partial \mu \right\|_\infty / 1 - \gamma \cdot \mu \right)^2}{\ell_{\nabla f}^3 \left( \ell_{\nabla f}^2 + \sigma^2 \right) \left( \ell_f / \ell_{\nabla f} + 1 \right)}$,*

$\eta_{\boldsymbol{\theta}}^{(t)} \asymp \dfrac{\varepsilon^8 \left( \left\| \partial \delta_\mu^{\boldsymbol{\pi}^*} / \partial \mu \right\|_\infty / 1 - \gamma \cdot \mu \right)^4}{\ell_{\nabla f}^5 \ell_f \left( \frac{\ell_f}{\ell_{\nabla f}}^2 + 1 \right)^4 \left( \ell_f^2 + \sigma^2 \right)^{3/2}} \wedge \dfrac{\varepsilon^2}{\ell_{\nabla f} \left( \ell_f^2 + \sigma^2 \right)}$, *then the best-iterate parameters and policies*

$(\boldsymbol{\theta}_{\text{best}}^{(T)}, \boldsymbol{x}_{\text{best}}^{(T)}) \in \arg\min_{t \in [T]} \left\| \nabla \widetilde{\varphi}(\boldsymbol{\theta}^{(t)}, \boldsymbol{x}^{(t)}) \right\|$ *converge to a stationary point of the exploitability,*
*i.e.,* $\left\| \nabla \widetilde{\varphi}(\boldsymbol{\theta}_{\text{best}}^{(T)}, \boldsymbol{x}_{\text{best}}^{(T)}) \right\| \leq \varepsilon$.

*Additionally, for any $\zeta, \xi \geq 0$ and for a sample size of equilibrium observations $\kappa \asymp 1/\varepsilon^2 \log(1/\zeta)$, with probability $1 - \zeta$, we have:*

$$\widehat{\varphi}(\boldsymbol{\theta}_{\text{best}}^{(T)}, \boldsymbol{x}_{\text{best}}^{(T)}) - \varphi(\boldsymbol{\theta}_{\text{best}}^{(T)}, \boldsymbol{x}_{\text{best}}^{(T)}) \leq \xi \tag{27}$$

*Proof of Theorem 3.* Although Daskalakis et al. (2020)'s Theorem 2 is stated for functions which are gradient-dominated-gradient-dominated, their proof falls through for any function which is non-convex-gradient-dominated. Define the *equilibrium distribution mismatch coefficient* $\|\partial \delta_\mu^{\boldsymbol{\pi}^\dagger} / \partial \mu\|_\infty$ as the Radon-Nikodym derivative of the state-visitation distribution of the Nash equilibrium $\boldsymbol{\pi}^\dagger$ w.r.t. the initial state distribution $\mu$. Under Assumption 3, by Corollary 1 and Theorem 4 of Bhandari & Russo (2019), we have that $g(\boldsymbol{\theta}, \boldsymbol{x}, \boldsymbol{y})$ is $\left( \|\partial \delta_\mu^{\boldsymbol{\pi}^\dagger} / \partial \mu\|_\infty / (1 - \gamma) \cdot \mu \right)$-gradient-dominated in $\boldsymbol{y}$ for all $\boldsymbol{\theta} \in \Theta$ and $\boldsymbol{x} \in \mathcal{X}$.

Moreover, according to analysis in Section 7.5, the variance of the gradient estimator is bounded. Hence, under our Theorem's assumptions, the assumptions of Theorem 2 of Daskalakis et al. (2020) are satisfied, and we have:

$$\frac{1}{T+1} \sum_{t=0}^{T} \left\| \nabla \widetilde{\varphi}(\boldsymbol{\theta}^{(t)}, \boldsymbol{x}^{(t)}) \right\| \leq \varepsilon \tag{28}$$

Taking a minimum across all $t = 0, 1, \ldots, T$, we then have:

$$\min_{t=0,1,\ldots,T} \left\| \nabla \widetilde{\varphi}(\boldsymbol{\theta}^{(t)}, \boldsymbol{x}^{(t)}) \right\| \leq \varepsilon \tag{29}$$

The second part is then a direct consequence of the hoeffding bound, whose assumptions are satisfied since the objective is bounded from above and from below by 0, as the objective is continuous and its domain is non-empty, and compact. $\qquad \square$

---

[10] We note that stationary points of the Moreau envelope correspond to stationary points of the subgradient of $\widehat{\varphi}$, but as exploitability is not neccessarily differentiable, the Moreau envelope is used to measure distance to a stationary point Lin et al. (2020).

## 7.3 RELATED WORK

**Microeconomics** The literature on characterizing agent preferences that can be rationalized by payoff functions, known under the names of *reveled preference theory* Samuelson (1948); Afriat (1967); Varian (1982; 2006) and the *integrability problem* Mas-Colell et al. (1995), far predates concerns of computing payoff functions that generate observed behavior. While revealed preference theory is concerned with understanding when a set of observed purchasing decisions for a consumer and associated market conditions (e.g., prices) is consistent with payoff-maximizing behavior, the integrability problem aims to characterize those consumption functions that can arise as the solution to a payoff maximization problem. The difference between revealed preference theory and the integrability problem is analogous to the difference between inverse optimization and inverse learning.

**Econometrics** A large body of work in the econometrics literature is dedicated to inverse game theory, with a recent focus on inferring bidders' valuations in online auctions. Nekipelov et al. 2015 analyzed inferring bidders' utilities in online ad auctions, assuming bidders are no-regret learners, and hence learn (coarse) correlated equilibrium. Syrgkanis et al. 2017 propose a method that infers agent types, assuming they play a Bayes-Nash equilibrium. More broadly, the identification literature Bresnahan & Reiss (1991); Lise (2001); Bajari et al. (2010) is closely related to our work, but usually does not address computational complexity concerns. Furthermore, the settings considered in this literature are overwhelmingly normal-form Bayesian, game, while our primary focus in this paper is (complete-information) stochastic games.

**Inverse Optimization.** Inverse optimization Heuberger (2004); Chan et al. (2021) seeks to recover the parameters of an optimization problem given access to the solution of the problem. One of the central results in inverse optimization demonstrates that one can recover the objective function of any linear inverse optimization problem from its solution by solving a linear program Chan et al. (2022). Our work considers the more general inverse problem for multiple agents with arbitrary objective, i.e., payoff functions, and solves it using a mathematical program as well.

**Inverse Algorithmic Game Theory** A literature that lies at the intersection of economics and computer science has aimed to provide computationally-efficient methods for rationalizing equilibria, but has mainly focused on specific types of games, such as matching Kalyanaraman & Umans (2008) and network formation games Kalyanaraman & Umans (2009). In the latter case, they showed that game attributes that are local to a player can be rationalized. More recently, Kuleshov & Schrijvers (2015) showed that correlated equilibria can be rationalized in polynomial-time in succint games. We note that these computational results concern stylized game models, and restrict certain aspects of the game, such as the size of the game's parameter space. In contrast, our results abstract away the issue of efficiently representatng the game's parameter space, and show that under appropriate parameterization, inverse equilbirium can be computed in polynomial time.

**Inverse Reinforcement Learning** Algorithms that infer the reward function of an agent operating within a Markov decision process Bellman (1952) have been studied extensively in recent years, starting with the initial investigations by Ng et al. (2000). These algorithms can be broadly categorized as maximum margin methods Ratliff et al. (2006); Silver et al. (2008); Abbeel & Ng (2004); Syed & Schapire (2007), i.e., methods that seek to maximize the margin between the value of observed behavior and the behavior associated with learned policy and rewards; maximum entropy methods Ziebart et al. (2008); Wulfmeier et al. (2015); Ziebart et al. (2008); Theodorou et al. (2010); Boularias et al. (2012; 2011), i.e., methods that maximize the entropy of the observed and the learned behaviors; Bayesian learning methods Ramachandran & Amir (2007); Choi & Kim (2011); Lopes et al. (2009); Levine et al. (2011); Babes et al. (2011), i.e., methods that learn a posterior distribution over parameters using Bayesian updating; and classification/regression methods Klein et al. (2012); Taskar et al. (2005); Klein et al. (2013); Brown et al. (2019), i.e., methods that learn parameters that minimize the distance between the observed behavior and behavior generated by learned behavior using the inferred parameters. Our methods, when used with only one player, can be characterized as a class of novel inverse reinforcement learning methods, which seek to recover parameters that minimize the players' regrets.

**Inverse Game Theory**   In multiagent settings, convex programming formulations have been proposed for inferring game parameters in normal-form games under the names of the inverse game theory problem Kuleshov & Schrijvers (2015) and the inverse equilibrium problem Waugh et al. (2013); Bestick et al. (2013). These methods focus on computing an inverse correlated equilibrium, and in the case of Waugh et al., further seek to reproduce observed equilibrium behavior via a maximum entropy correlated equilibrium. Hadfield-Menell et al. 2016 consider cooperative inverse reinforcement learning, which can be seen as an inverse Nash equilibrium problem in a particular zero-sum imperfect-information game, but this method is not accompanied by computational guarantees.

**Multiagent Inverse Reinforcement Learning**   Multiagent inverse reinforcement learning generalizes inverse game theory from normal-form games to Markov games Natarajan et al. (2010); Lin et al. (2019); Yu et al. (2019); Lin et al. (2017); Fu et al. (2021). Even more importantly, instead of observing equilibrium policies, only sample trajectories from equilibrium policies are observed. Lin et al. (2019) study the inverse Nash equilibrium problem in zero-sum games Lin et al. (2017), and extend their methods to solve for inverse correlated equilbrium in general-sum stochastic games, and inverse Nash equilibrium in a restricted class of adversarial stochastic games. Yu et al. (2019) propose gradient-based algorithms for computing inverse quantal response equilibria with function approximation.

## 7.4 MARKETS EXPERIMENTS

### 7.4.1 STATIC FISHER MARKETS

A *(one-shot) Fisher market* consists of $n$ buyers and $m$ divisible goods with unit supply(Brainard et al., 2000). Each buyer $i \in [n]$ is endowed with a budget $b_i \in \mathcal{B}_i \subseteq \mathbb{R}_+$ and a utility function $u_i : \mathbb{R}_+^m \times \mathcal{T}_i \to \mathbb{R}$, which is parameterized by a type $t_i \in \mathcal{T}_i$ that defines a preference relation over the consumption space $\mathbb{R}_+^m$. An instance of a Fisher market is then a tuple $\mathcal{M} \doteq (n, m, \boldsymbol{u}, \boldsymbol{t}, \boldsymbol{b})$, where $\boldsymbol{u} \doteq (u_1, \dots, u_n)$ is a vector-valued function of all utility functions and $\boldsymbol{b} \doteq (b_1, \dots, b_n) \in \mathbb{R}_+^n$ is the vector of buyer budgets. When clear from context, we simply denote $\mathcal{M}$ by $(\boldsymbol{t}, \boldsymbol{b})$.

Given a Fisher market $(\boldsymbol{t}, \boldsymbol{b})$, an *allocation* $\boldsymbol{X} = (\boldsymbol{x}_1, \dots, \boldsymbol{x}_n)^T \in \mathbb{R}_+^{n \times m}$ is a map from goods to buyers, represented as a matrix, s.t. $x_{ij} \geq 0$ denotes the amount of good $j \in [m]$ allocated to buyer $i \in [n]$. Goods are assigned *prices* $\boldsymbol{p} = (p_1, \dots, p_m)^T \in \mathbb{R}_+^m$. A tuple $(\boldsymbol{X}^*, \boldsymbol{p}^*)$ is said to be a *competitive equilibrium (CE)* (Arrow & Debreu, 1954; Walras, 1896) if 1. buyers are utility maximizing, constrained by their budget, i.e., $\forall i \in [n], \boldsymbol{x}_i^* \in \arg\max_{\boldsymbol{x} : \boldsymbol{x} \cdot \boldsymbol{p}^* \leq b_i} u_i(\boldsymbol{x}, \boldsymbol{t}_i)$; and 2. the market clears, i.e., $\forall j \in [m], p_j^* > 0 \Rightarrow \sum_{i \in [n]} x_{ij}^* = e_j$ and $p_j^* = 0 \Rightarrow \sum_{i \in [n]} x_{ij}^* \leq e_j$.

The set of CE of any Fisher market $(\boldsymbol{t}, \boldsymbol{b})$ with continuous, concave, and homogeneous[11] utility functions is equal to the set of Nash equilibria of the *Eisenberg-Gale min-max game*,[12] a convex-concave min-max game between a seller who chooses prices $\boldsymbol{p} \in \mathbb{R}_+^m$ and buyers who collectively choose allocations $\boldsymbol{X} \in \mathbb{R}_+^{n \times m}$: the objective function of this game comprises two sums: the first is the logarithmic Nash social welfare of the buyers' utility , while the second is the profit of a fictional auctioneer who sells the goods in the market:

$$\min_{\boldsymbol{p} \in \mathbb{R}_+^m} \max_{\boldsymbol{X} \in \mathbb{R}_+^{n \times m}} f(\boldsymbol{p}, \boldsymbol{X}; \boldsymbol{t}, \boldsymbol{b}) \doteq \sum_{i \in [n]} b_i \log\left(u_i(\boldsymbol{x}_i, \boldsymbol{t}_i)\right) + \sum_{j \in [m]} \left( p_j - p_j \sum_{i \in [n]} x_{ij} \right) \quad (30)$$

Therefore, for any Fisher market $\mathcal{M} \doteq (n, m, \boldsymbol{u}, \boldsymbol{t}^\dagger, \boldsymbol{b}^\dagger)$, we can construct a inverse game $\mathcal{G}^{-1} \doteq (\mathcal{G}^{\boldsymbol{\theta}^\dagger} / \boldsymbol{\theta}^\dagger, \boldsymbol{x}^\dagger)$ where $\mathcal{G}^{\boldsymbol{\theta}^\dagger}$ is the corresponding Eisenberg-Gale min-max game (Equation (30)) parameterized by the true types and budgets $\boldsymbol{\theta}^\dagger = (\boldsymbol{t}^\dagger, \boldsymbol{b}^\dagger)$, and $\boldsymbol{x}^\dagger = (\boldsymbol{X}^*, \boldsymbol{p}^*)$ is not only a NE of the game $\mathcal{G}^{\boldsymbol{\theta}^\dagger}$ but also a CE of market $\mathcal{M}$. Our goal is to recover the true market parameters $\boldsymbol{\theta}^\dagger = (\boldsymbol{t}^\dagger, \boldsymbol{b}^\dagger)$ given the observed $\boldsymbol{x}^\dagger = (\boldsymbol{X}^*, \boldsymbol{p}^*)$, by solving this inverse game problem using Theorem 3.1 and Algorithm 1.

We ran two different experiments[13]. First, we solved a simpler inverse game problem where the true type $\boldsymbol{t}^\dagger$ is given, and we just need to retrieve the true budgets $\boldsymbol{b}^\dagger$; then, we attempt to recover both true type and true budgets simultaneously. For both experiments, we created 500 markets with each of these three (standard) classes of utility functions parameterized by types: 1. *linear*: $u_i(\boldsymbol{x}_i; \boldsymbol{t}_i) = \sum_{j \in [m]} t_{ij} x_{ij}$; 2. *Cobb-Douglas (CD)*: $u_i(\boldsymbol{x}_i; \boldsymbol{t}_i) = \prod_{j \in [m]} x_{ij}^{t_{ij}}$; and 3. *Leontief*: $u_i(\boldsymbol{x}_i; \boldsymbol{t}_i) = \min_{j \in [m]} \left\{ \frac{x_{ij}}{t_{ij}} \right\}$. Then, we ran Algorithm 1 on min-max optimization problem defined in Equation (1) to compute the inverse Nash Equilibrium of the inverse game $\mathcal{G}^{-1}$ defined above. Finally, for each utility type, we recorded the percentage of markets that we could recover the true parameters, i.e., the markets for which our computed parameters is close enough to the true parameters, and the average exploitability of the observed equilibrium evaluated under the computed parameters across all markets.

Table 2 shows that when only retrieving budgets,we were able to recover all the parameters and minimize exploitability in markets with Linear and Cobb-Douglas utilities, but we hardly do so in Leontief markets. The difficulty in this case likely arises from two aspects: first, Leontief utility function is not differentiable, so the min-max optimization problems associated to Leontief markets are not smooth; moreover, for any Leontief Fisher market, the CE is not guaranteed to be unique. When computing budgets and types at once, while our algorithm can still minimize exploitability for

---

[11]A function $f : \mathbb{R}^m \to \mathbb{R}$ is called *homogeneous of degree $k$* if $\forall \boldsymbol{x} \in \mathbb{R}^m, \lambda > 0, f(\lambda \boldsymbol{x}) = \lambda^k f(\boldsymbol{x})$.

[12]This min-max game corresponds to the Lagrangian saddle-point formulation of the Eisenberg-Gale program Gale (1989); Jain et al. (2005).

[13]We include a detailed description of our experimental setup in the appendix.

both Linear nad Cobb-Douglas markets, it cannot really retrive true parameters in Linear markets. This may due to the fact that, in Linear markets, though competitive equilibrium prices are unique, the competitive allocations are not guaranteed to be unique; by contrast, the CEs are always unique in Leontief markets.

**Hyperparameters**  We randomly initialized 500 different linear, Leontief, Cobb-Douglas Fisher markets, each with 3 buyers and 2 goods. Buyer $i$'s budget $b_i$ was drawn randomly from a uniform distribution ranging from 0 to 10 (i.e., $U[0, 10]$), and each buyer $i$'s type for good $j$, $t_{ij}$, was drawn randomly from $U[0, 10]$.

For Fisher markets with all three class of utilities, we ran our algorithm for 5000 iterations with learning rate $\eta_\theta = 0.01$. Moreover, we stop our algorithm when our computed inverse Nash equilibrium $\theta$ is closed enough to the original parameter $\theta^\dagger$: $||\frac{\theta - \theta^\dagger}{\theta^\dagger}||_2 \le \epsilon$ where we set $\epsilon = 0.1$.

### 7.4.2 COURNOT COMPETITION AND BERTRAND COMPETITION

A *Cournot competition model* $\mathcal{C} \doteq (n, c, P)$ consists of $n$ firms that produce a homogeneous product, and each firm $i$ chooses a quantity level of production $q_i$ that maximizes its profits. All firms face a marginal cost $c$. That is, for a given firm $i$, the cost of producing $q_i$ unit of good is $cq_i$. The price function $P$ takes the total production of all firms $Q_{total} = \sum_{i \in [n]} q_i$ as input and outputs the unit prices for the good. Thus, the profit function for firm $i$ is $f_i(q_i, q_{-i}; c) = q_i(P(\sum_{i \in [n]} q_i) - c)$. $\mathbf{Q}^*$ is a Nash equilibria of the Cournot game if and only if $q_i^* \in \arg\max_{q_i in \mathbb{R}_+} f_i(q_i, q_{-i}^*; c)$ for all $i \in [n]$.

A *Bertrand competition model* $\mathcal{B} \doteq (n, c, D)$ is also a competition model that consists of $n$ firms producing a homogeneous product, but this time, each firm $i$ set prices $p_i$ to maximize its profits. All firms face a marginal cost $c$. That is, for a given firm $i$, the cost of producing $q_i$ unit of good is $cq_i$. The demand function $D$ takes the minimum price proposed by the firms $p_{\min} = \min_{i \in [n]} p_i$ as input and outputs the demand for that good in the whole market. Firm $i$'s individual demand function is a function of the price set by each firm:

$$D_i(p_i, p_{-i}) = \begin{cases} D(p_{\min}) & p_i = p_{\min}, p_j \ge p_{\min} \forall j \neq i \\ \frac{D(p_{\min})}{n} & p_i = p_{\min}, n = \# \text{ of } j \in [n] \text{ with } p_j = p_{\min} \\ 0 & p_i \neq p_{\min} \end{cases} \quad (31)$$

Thus, the profit function for the firm $\boldsymbol{i}$ is $f_i(p_i, p_{-i}; c) = D_i(p_i, p_{-i})(p_i - c)$. $\boldsymbol{p}^*$ is a Nash equilibria of the Bertrand game if and only if $p_i^* \in \arg\max_{p_i in \mathbb{R}_+} f_i(p_i, p_{-i}^*; c)$ for all $i \in [n]$.

In experiments, we generated 500 Cournot competition models and 500 Bertrand competition models and attempted to retrieve their true parameter, i.e., marginal costs, given equilibrium productions/equilibrium prices respectively. Table 2 shows that our algorithm can effectively recover the true parameters in Cournot games and minimize the exploitability of the observed equilibrium evaluated under the computed parameters. In the Bertrand games, though we could only recover $78\%$ true parameters, the average exploitability of the observed equilibrium evaluated under the computed inverse Nash equilibrium is mostly minimized.

**Hyperparameters**  We randomly initialized 500 different duopoly Cournot competitions and duopoly Bertrand competitions. Marginal costs was drawn randomly from a uniform distribution ranging from 2 to 20 (i.e., $U[2, 20]$) for both Cournot and Bertrand. Moreover, we define the price functions in Cournot competitions as $P(q_{total}) = a + bq_{total}$ where $a \sim U[10, 100]$ and $b \sim U[-10, -0.01]$. We define the demand functions in Bertrand competitions as $D(p_{\min}) = c + dp_{\min}$ where $c \sim U[10, 100]$ and $d \sim U[-10, -0.01]$.

For Cournot competitions, we ran our algorithm for 10000 iterations with learning rate $\eta_\theta = 0.01$, and for Bertrand competitions, we ran our algorithm for 250 iterations with learning rate $\eta_\theta = 0.3$. Moreover, we stop our algorithm when our computed inverse Nash equilibrium $\theta$ is closed enough to the original parameter $\theta^\dagger$: $||\frac{\theta - \theta^\dagger}{\theta^\dagger}||_2 \le \epsilon$ where we set $\epsilon = 0.1$.

### 7.4.3 STOCHASTIC FISHER MARKETS

*A (static) Fisher market* $(n, m, \mathcal{C}, \boldsymbol{u}, \boldsymbol{e}, \boldsymbol{t}, \boldsymbol{b})$, $(\boldsymbol{e}, \boldsymbol{t}, \boldsymbol{b})$ when clear from context, consists of $n \in \mathbb{N}_{++}$ buyers and $m \in \mathbb{N}_{++}$ divisible goods Brainard et al. (2000). Each buyer $i \in [n]$ is represented by a tuple $(\mathcal{C}_i, u_i, \boldsymbol{t}_i, b_i)$, $(\boldsymbol{t}_i, b_i)$ when clear from context, which consists of a *budget* $b_i \in \mathcal{B}_i \subseteq \mathbb{R}_+$ of some numéraire good it is endowed with, a *utility function* $u_i : \mathcal{C}_i \times \mathcal{T}_i \to \mathbb{R}_+$, which is parameterized by a *type* $\boldsymbol{t}_i \in \mathcal{T}_i$ s.t. $u_i(\,\cdot\,; \boldsymbol{t}_i)$ defines a preference relation over the *consumption space* $\mathcal{C}_i \subseteq \mathbb{R}_+^m$. Each good is characterized by a supply $e_j \in \mathcal{E}_j \subseteq \mathbb{R}_+$. We denote the collection of all utility functions $\boldsymbol{u} \doteq (u_1, \ldots, u_n)$, the collection of buyer types $\boldsymbol{t} \doteq (\boldsymbol{t}_1, \ldots, \boldsymbol{t}_n)$, the collection of buyer budgets $\boldsymbol{b} \doteq (b_1, \ldots, b_n) \in \mathbb{R}_+^n$, the collection of all good supplies $\boldsymbol{e} \doteq (e_1, \ldots, e_m) \in \mathbb{R}_+^m$, the joint space of consumptions $\mathcal{C} \doteq \bigtimes_{i \in [n]} \mathcal{C}_i$, the joint space of types $\mathcal{T} \doteq \bigtimes_{i \in [n]} \mathcal{T}_i$, and the joint space of budgets $\mathcal{B} \doteq \bigtimes_{i \in [n]} \mathcal{B}_i$.

**Definition 1** (Stochastic Fisher Market Game). *Given a stochastic Fisher market $\mathcal{F} \doteq (n, m, l, \mathcal{S}, \boldsymbol{u}, p, \gamma, \mu)$, we define the Stochastic Fisher Market Game $\mathcal{M}^{\mathcal{F}} \doteq (n, \mathcal{S}, \mathcal{A}, \mathcal{B}, \boldsymbol{r}, p, \gamma, \mu)$ where:*

$$\mathcal{S} \doteq \mathcal{O} \times \mathcal{E} \times \mathcal{B} \times \mathcal{T} \qquad \mathcal{A} \doteq \mathcal{P} \qquad \mathcal{B}(\boldsymbol{s}) \doteq \mathcal{C} \times \mathcal{A}(\boldsymbol{s}) \tag{32}$$

$$\boldsymbol{r}((\boldsymbol{\omega}, \boldsymbol{e}, \boldsymbol{t}, \boldsymbol{b}), \boldsymbol{p}, (\boldsymbol{X}, \boldsymbol{\beta})) \doteq \boldsymbol{p} \cdot \left( \boldsymbol{e} - \sum_{i \in [n]} \boldsymbol{x}_i \right) + \sum_{i \in [n]} (b_i + \beta_i) \log \left( \frac{u_i(\boldsymbol{x}_i; \boldsymbol{t}_i)}{b_i + \beta_i} \right) \tag{33}$$

**Experimental setup**   We use Jax, and Haiku to traint he simulacrum policies and use a feedforward neural network with 4 layers with 200 nodes. We run our experiments with 5 random seed and report the best results.

## 7.5 GRADIENT ESTIMATORS

Notice that the deterministic policy gradient theorem tells us that to compute a the policy gradient we need to compute the gradient of state-action value function with respect to the actions and then multiply it by the gradient of the policy w.r.t. the policies. Since we have access to a first order oracle of the reward and transition we can then compute gradient of the cumulative regret with the following quantities:

$$
f_{\boldsymbol{x}}(\boldsymbol{\theta}, \boldsymbol{y}; \boldsymbol{H}, \boldsymbol{h}')
$$
$$
\propto \sum_{i \in [n]} \left[ \nabla_{\boldsymbol{a}} r_i(\boldsymbol{s}^{i,(0)}, \boldsymbol{a}^{i,(0)}; \boldsymbol{\theta}) \right.
$$
$$
\left. + \gamma \nabla_{\boldsymbol{a}} p(\boldsymbol{s}^{i,(1)} \mid \boldsymbol{s}^{i,(0)}, \boldsymbol{a}^{i,(0)}) \left[ r_i\left(\boldsymbol{s}^{i,(1)}, \boldsymbol{a}^{i,(1)}; \boldsymbol{\theta}\right) + \sum_{t=2}^{\infty} r_i\left(\boldsymbol{s}^{i,(t)}, \boldsymbol{a}^{i,(t)}\right) \prod_{k=2}^{t} \gamma^{k+1} p(\boldsymbol{s}^{i,(k)} \mid \boldsymbol{s}^{i,(k-1)}, \boldsymbol{a}^{i,(k-1)}) \right] \right]
$$

$$(34)$$

$$
f_{\boldsymbol{\theta}}(\boldsymbol{\theta}, \boldsymbol{y}; \boldsymbol{h}, \boldsymbol{h}') \doteq \sum_{i \in [n]} \left[ \sum_t \nabla_{\boldsymbol{\theta}} r_i(\boldsymbol{s}^{i,(t)}, \boldsymbol{a}^{i,(t)}; \boldsymbol{\theta}) - \sum_t \nabla_{\boldsymbol{\theta}} r_i(\boldsymbol{s}'^{(t)}, \boldsymbol{a}'^{(t)}; \boldsymbol{\theta}) \right] \quad (35)
$$

Under Assumption 3, these estimators are unbiased estimates of the gradients $\nabla_{\boldsymbol{\theta}} f$ and $\nabla_{\boldsymbol{x}} f$, respectively. Assuming these estimators have bounded variance, we can now solve the min-max optimization problem $\min_{\boldsymbol{\theta} \in \Theta} \max_{\boldsymbol{x} \in \mathcal{X}} f(\boldsymbol{\theta}, \boldsymbol{x})$ for an $\varepsilon$-inverse NE in $O(1/\varepsilon)$ iterations via stochastic gradient descent (Algorithm 2).[14] It thus remains to show that the variance of the gradient estimators are bounded: i.e., there exists $\sigma \in [0, \infty)$ s.t. for all $(\boldsymbol{\theta}, \boldsymbol{x}) \in \Theta \times \mathcal{X}$, $\left\| \mathbb{E}_{\boldsymbol{h}, \boldsymbol{h}'}[(f_{\boldsymbol{\theta}}, f_{\boldsymbol{x}})(\boldsymbol{\theta}, \boldsymbol{x}; \boldsymbol{h}, \boldsymbol{h}')] - \nabla f(\boldsymbol{\theta}, \boldsymbol{x}) \right\| \leq \sigma$. Since rewards, transitions, and policies, are twice continuously-differentiable, both the gradient estimates $(f_{\boldsymbol{\theta}}, f_{\boldsymbol{x}})$ and $\nabla f$ also are, and we have: $\left\| \mathbb{E}_{\boldsymbol{h}, \boldsymbol{h}'}[(f_{\boldsymbol{\theta}}, f_{\boldsymbol{x}})(\boldsymbol{\theta}, \boldsymbol{x}; \boldsymbol{h}, \boldsymbol{h}')] - \nabla f(\boldsymbol{\theta}, \boldsymbol{x}) \right\| \leq \max_{\boldsymbol{\theta}, \boldsymbol{x}, \boldsymbol{h}, \boldsymbol{h}'} \left\| (f_{\boldsymbol{\theta}}, f_{\boldsymbol{x}})(\boldsymbol{\theta}, \boldsymbol{x}; \boldsymbol{h}, \boldsymbol{h}') \right\| = \left\| (f_{\boldsymbol{\theta}}, f_{\boldsymbol{x}}) \right\|_{\infty}$, where the max is well defined, since the objective is continuous and the maximization domains $\mathcal{S}, \mathcal{A}, \mathcal{X}, \Theta$ are non-empty and compact. This means that, under Assumption 3, the variance of gradient estimator $(f_{\boldsymbol{\theta}}, f_{\boldsymbol{x}})$ is bounded by $\sigma^2 \doteq \left\| (f_{\boldsymbol{\theta}}, f_{\boldsymbol{x}}) \right\|_{\infty}^2$.

---

[14] With additional care, the assumption made on the rewards and probability transition functions in Part 2 of Assumption 3 can be weakened to continuous differentiability and local Lipschitz-continuity, respectively (see Lemma 3.2 of Suh et al. (2022)) to obtain unbiased estimates; for clarity we make the stronger assumption.