# OpenReview forum: "Efficient Inverse Multiagent Learning"
_ICLR.cc/2024/Conference — ICLR 2024 spotlight_

### Official Review · Reviewer_zp62 · 2023-10-30

**Soundness:** 4 excellent
**Presentation:** 3 good
**Contribution:** 3 good
**Rating:** 8
**Confidence:** 3

**Summary:**

Game theory provides a structured approach to predicting outcomes of interactions between rational agents. Inverse game theory deals with situations where the players' behavioral models are unknown and aims to deduce the payoff functions that explain observed actions as equilibria. This paper presents a new approach for solving inverse equilibrium problems in a range of games, using generative adversarial optimization to match game-theoretic models to observed data and make predictions, as exemplified by modeling the Spanish electricity market.

**Strengths:**

The authors study an interesting and challenging problem of inverse MARL. The theoretical results are nice; they are simple yet relevant and impactful.
The experiments on an electricity market are well thought and designed.

**Weaknesses:**

The readability can be improved. I think Section 3 never mentions that it is for the one-shot game setting.
There is no methodological contributions. All presented algorithms are simple extension of known algorithms. (On the other hand we should not invent/propose algorithms just for the sake of proposing them).

**Questions:**

Minor remark/question: what is the meaning of that weird symbol in Theorems 3.2, 4.1, 5.2? I assume it means of the same order as (but I don't think this is standard notation.)
In Theorem 3.2, it is a little bit surprising that the optimal is obtained by averaging prior solutions instead of the last one. What is the intuition behind averaging (which includes initial solutions that can be of very low quality)? I assume the proof is correct.

---

> ### Author Response · Authors · 2023-11-17
>
> Thank you for your review!
>
> ### Weaknesses
>
> > **W1)**  The readability can be improved. I think Section 3 never mentions that it is for the one-shot game setting. There is no methodological contributions.
>
> **Response to W1)**: In the second paragraph of Section 3, we define an inverse game as a tuple comprising a one-shot parametric game whose parameter values are missing, together with a Nash equilibrium action profile.
>
> > **W2)** All presented algorithms are simple extension of known algorithms. (On the other hand we should not invent/propose algorithms just for the sake of proposing them).
>
> **Response to W2)** As you mention, we have not developed any groundbreaking methodologies, but have simply introduced a novel mathematical characterization of a longstanding problem. Perhaps the strength of our paper lies in its simplicity.
>
> ### Questions
>
> > (Q1) Minor remark/question: what is the meaning of that weird symbol in Theorems 3.2, 4.1, 5.2? I assume it means of the same order as (but I don't think this is standard notation.)
>
> We believe you are referring to the notation $\eta_{\boldsymbol{y}}^{(t)} \asymp \varepsilon^4$. This notation is equivalent to big theta notation, i.e., $\eta_{\boldsymbol{y}}^{(t)} \in \Theta(\varepsilon^4)$. An example of the use of this notation can be found in [1]. Regardless, we will add a footnote to clarify.
>
> > (Q2) In Theorem 3.2, it is a little bit surprising that the optimal is obtained by averaging prior solutions instead of the last one. What is the intuition behind averaging (which includes initial solutions that can be of very low quality)? I assume the proof is correct.
>
> **Response to Question**: Simple gradient descent ascent methods are not guaranteed to converge in last-iterates in min-max optimization problems, and can even diverge (see, for instance, Mertikopoulos et al. [2]). A common remedy is to average the iterates. As we mention in the sentence preceeding Theorem 3.2, we can instead obtain last-iterate convergence using extragradient descent ascent or optimistic gradient descent ascent.
>
> **References**
>
> [1] Daskalakis, Constantinos, Dylan J. Foster, and Noah Golowich. "Independent policy gradient methods for competitive reinforcement learning." Advances in neural information processing systems 33 (2020): 5527-5540.
>
> [2] Panayotis Mertikopoulos, Christos Papadimitriou, and Georgios Piliouras. Cycles in adversarial regularized learning. In Proceedings of the Twenty-Ninth Annual ACM-SIAM Symposium on Discrete Algorithms, pages 2703–2717, 2018.

---

### Official Review · Reviewer_F6LX · 2023-11-01

**Soundness:** 4 excellent
**Presentation:** 3 good
**Contribution:** 4 excellent
**Rating:** 10
**Confidence:** 3

**Summary:**

This paper formalizes the problem of inverse game-theory (determining
equilibrium strategies and associated game structure from historical play) as a
novel min-max optimization problem and solves it using a primal-dual gradient
method. This technique is deployed on a practical and interesting application
domain of market pricing dynamics.

**Strengths:**

The simple formulation of the set of inverse Nash equilibria (NE) as a min-max
game is elegant and appears to be original. If it is indeed original, for this
alone, the paper merits publication and should be highlighted.

The paper overall is well written and showcases immediate applications of the
proposed solution to an important and practical domain as a proof-of-concept. I
believe these results are significant and will be impactful.

**Weaknesses:**

As an easily rectified issue, Figure 1 could have been better represented by
plotting residuals over time or, by subsampling the data, plotting mean
residuals with error bars.

As a minor complaint, I do not prefer the language of "generative-adversarial"
(especially not in terms of a "discriminator"), even if this is the closest
analogy familiar to machine learning practitioners: This is a standard min-max
optimization problem that need not be wed to the ML setting.

**Questions:**

Remark 1 is indeed interesting, but it is not obvious. Did I miss an associated
proof or example?

Why was the proof of Theorem 3.2 omitted? Was it rephrased to appear as Theorem
6.1 in the supplementary material?  Establishing the convergence rates of
various algorithms is not my expertise, but the results seem reasonable,
especially given assumptions of convexity and Lipshitz smoothness.

---

> ### Author Response · Authors · 2023-11-17
>
> Thank you for your kind and encouraging review, it means a lot to us!
>
> ### Weaknesses
>
> >**W1)** As an easily rectified issue, Figure 1 could have been better represented by plotting residuals over time or, by subsampling the data, plotting mean residuals with error bars.
>
> **Response to W1)**: This is a great suggestion. Thank you. We will indeed plot residuals over time and error bars over the five seeds for which we ran our experiments.
>
> >**W2)** As a minor complaint, I do not prefer the language of "generative-adversarial" (especially not in terms of a "discriminator"), even if this is the closest analogy familiar to machine learning practitioners: This is a standard min-max optimization problem that need not be wed to the ML setting.
>
> **Response to W2)**:
>
> Thank you for this feedback. The "generative" language is meant to allude to the generative model fitting problem within the larger technical development of our approach. That said, the discriminator language is not quite right. We will give further thought to this concern, perhaps by replacing "generative-adversarial" simply by "min-max".
>
> ### Questions
>
> >**Q1**: Remark 1 is indeed interesting, but it is not obvious. Did I miss an associated proof or example?
>
> **Response to Q1**: This remark follows from a folklore theorem, which states that set of solutions to convex-concave min-max optimization problems (i.e., min-max problems in which the objective is convex-concave, and the constraints are non-empty, compact, and convex) are convex. This fact can be seen as a corollary of the set of Nash equilibria being convex in zero-sum, potential, and monotone games. You can find a proof in Nau et al [1]; for completeness we also provide a proof here.
>
> Consider a convex-concave min-max optimization problem $\min_{\mathbf{x} \in \mathcal{X}} \max_{\mathbf{y} \in \mathcal{Y}} f(\mathbf{x},\mathbf{y})$ where $f: \mathcal{X} \times \mathcal{Y} \to \mathbb{R}$ is convex-concave and $\mathcal{X}, \mathcal{Y}$ are non-empty, compact, and convex sets. Let $V(\mathbf{x}) \doteq \max_{\mathbf{y} \in \mathcal{Y}} f(\mathbf{x},\mathbf{y})$. By Danskin's theorem [2], $V$ is convex since it is the maximum of a set of convex functions. Hence, by Theorem 2.6 of Rockafeller and Wets [3], the set of solutions $\arg \min_{\mathbf{x} \in \mathcal{X}} V(\mathbf{x}) = \arg\min_{\mathbf{x} \in \mathcal{X}} \max_{\mathbf{y} \in \mathcal{Y}} f(\mathbf{x},\mathbf{y})$ is convex.
>
> >**Q2**: Why was the proof of Theorem 3.2 omitted? Was it rephrased to appear as Theorem 6.1 in the supplementary material? Establishing the convergence rates of various algorithms is not my expertise, but the results seem reasonable, especially given assumptions of convexity and Lipschitz smoothness.
>
> **Response to Q2**: We refer you to our common answer.
>
> **References**
>
> [1] Nau, Robert, Sabrina Gomez Canovas, and Pierre Hansen. "On the geometry of Nash equilibria and correlated equilibria." International Journal of Game Theory 32 (2004): 443-453.
>
> [2] Danskin, John M. "The theory of max-min, with applications." SIAM Journal on Applied Mathematics 14.4 (1966): 641-664.
>
> [3] R Tyrrell Rockafellar and Roger J-B Wets. Variational analysis, volume 317. Springer Science and Business Media,2009

---

### Official Review · Reviewer_Wrco · 2023-11-01

**Soundness:** 2 fair
**Presentation:** 1 poor
**Contribution:** 2 fair
**Rating:** 6
**Confidence:** 2

**Summary:**

This paper studied inverse game theory to find parameters of the payoff functions of the game. Polynomial time and sample efficient algorithms are provided and claimed.

**Strengths:**

1. This paper formulate the inverse game as an generative-adversarial optimization problem and provide polynomial time algorithms.

**Weaknesses:**

1. The proofs are not completed, e.g., I cannot find the proofs for Theorem 4.1 and Theorem 5.2.

2. The presentation can be further improved, e.g., more intuitions about the assumptions and theorems.

**Questions:**

1. See weaknesses.

2. Can you further polish the paper? Some typos: for example, in the fifth line of the abstract, should it be "to solve them"?

---

> ### Author Response · Authors · 2023-11-17
>
> Thank you for your review!
>
> ### Weaknesses
>
> >**W1)** The proofs are not completed, e.g., I cannot find the proofs for Theorem 4.1 and Theorem 5.2.
>
> **Response to W1)**:  We refer you to our common answer.
>
> >**W2)** The presentation can be further improved, e.g., more intuitions about the assumptions and theorems.
>
> **Response to W2)**: We will try to add additional intuition to the main paper, beyond what already appears at the bottom of page 6 and top of page 7 (for example), respecting the space constraints. You can also already find some additional discussion in the appendix (see page 16).
>
> ### Questions
>
> >**Q2)** Can you further polish the paper? Some typos: for example, in the fifth line of the abstract, should it be "to solve them"?
>
> **Response to Q2)**: We will do our best to thoroughly proofread the paper again to correct all grammatical, semantic, and syntactical errors in the camera-ready version.

---

> > ### Author Response · Authors · 2023-11-21
> >
> > Dear reviewer,
> >
> > Since the end of the discussion period is nearing, we were wondering whether you have further questions. We would like to have the opportunity to discuss and resolve potential further inquiries.
> >
> > Thanks in advance,
> > The authors

---

> > > ### Comment · Reviewer_Wrco · 2023-11-22
> > >
> > > Thanks for the response. Similar to reviewer WrBZ, the proofs are still a little hard for me to follow. I am sorry that I may not be able to check the detailed technique. However, I agree with the strengths pointed out by other reviewers. I have increased my rate to a 6.

---

> > > > ### Author Response · Authors · 2023-11-23
> > > >
> > > > Thank you for your time and answer!

---

### Official Review · Reviewer_WrBZ · 2023-11-08

**Soundness:** 2 fair
**Presentation:** 2 fair
**Contribution:** 3 good
**Rating:** 6
**Confidence:** 2

**Summary:**

The paper introduces a generative-adversarial (or min-max) characterization of the inverse game theory problem where a generator provides payoff parameters that minimize total regret, and a discriminator looks for action-profiles that maximize it. A min-max objective mimicking this two-player game is optimized to estimate the inverse equilibrium using gradient descent ascent algorithm (and other variations of it). It is further proposed that for games that satisfy certain assumptions guaranteeing convex-concavity of the objective, the algorithm converges in a number of iterations that are polynomial in the precision of the obtained inverse equilibrium. This formulation is further extended to give algorithms for multi-agent inverse reinforcement learning and multi-agent apprenticeship learning, accompanied by polynomial time (and space) convergence guarantees under appropriate assumptions. Experiments are conducted to identify categories of games for which the method is effective, and whether its usefulness goes beyond provided theoretical limits.

**Strengths:**

* An inverse game theoretic perspective to multi-agent inverse reinforcement learning is certainly a novel direction to approach the problem with. Backed by results in inverse game theory, this approach leads to algorithms with desirable convergence guarantees that prior work in multi-agent imitation learning does not provide.

* The low restrictiveness of the assumptions made allow for the framework to be effective on a vast majority of markov games, leading to useful and efficient solutions on a wide variety of multi-agent problems.

* While the paper focuses on the inverse nash equilibrium, the simplicity of the objective allows for easy extensions of the framework to alternative game theory solution concepts.

* All presented algorithms are succinct and easy to understand. Sufficient mathematical background is provided as and when necessary.

**Weaknesses:**

* It would be helpful to expand on the proofs of theorems 6.1, 6.2, and 6.3 in the supplementary material. I know that a reference has been provided, but a slight explanation of the cited result and how it relates to the theorem in question would be nice.

* Although a comparison of the method has been shown with the ARIMA model on the spanish electricity market data, it would be beneficial to have a comparison with prior methods in inverse multi-agent reinforcement learning. Especially in terms of efficiency since it's one of the main points of the paper. The abstract says that the method outperforms other widely-used methods (plural), and we only get to see it being compared with one other model which is specific to time-series data.

* Some comparison/contextualization with prior work in multiagent inverse reinforcement learning would also be helpful.

**Questions:**

What does the term $\psi(\pi, \rho; \theta)$ in the "Multiagent Apprenticeship Learning" section expand to? Cannot seem to find a definition anywhere.

Assuming that Algorithm 3 was used on the spanish electricity market data, how was the observation distribution specified?

---

> ### Author Response · Authors · 2023-11-17
>
> Thank you for your review!
>
> ### Weaknesses
>
> >**W1)** $\bullet$ It would be helpful to expand on the proofs of theorems 6.1, 6.2, and 6.3 in the supplementary material. I know that a reference has been provided, but a slight explanation of the cited result and how it relates to the theorem in question would be nice.
>
> **Response to W1)**: We refer you to our common answer.
>
> >**W2)** $\bullet$ Although a comparison of the method has been shown with the ARIMA model on the spanish electricity market data, it would be beneficial to have a comparison with prior methods in inverse multi-agent reinforcement learning. Especially in terms of efficiency since it's one of the main points of the paper. The abstract says that the method outperforms other widely-used methods (plural), and we only get to see it being compared with one other model which is specific to time-series data.
>
> **Response to W2)**: To our knowledge, all existing methods of inverse multiagent reinforment learning apply only to finite state and action Markov games, while our electricity market model is a continuous state and action Markov game. As a result, comparing to these methods would require potentially non-trivial extensions. Instead, we chose to compare our method to a statistical method, namely that of ARIMA, only. We will adjust our language to correct the plural.
>
> >**W3)** $\bullet$ Some comparison/contextualization with prior work in multiagent inverse reinforcement learning would also be helpful.
>
>
> **Response to W3)**: A more extensive related works section is included in the third section of the appendix. There, we summarize related work not only in inverse multiagent reinforcement learning, but also in microeconomics, econometrics, and algorithmic game theory. Thank you for pointing out that a reference to this more detailed related work section is missing. We will be sure to correct this oversight in the camera-ready version.
>
> ### Questions
>
> >**Q1) ** $\bullet$ What does the term $\psi(\pi,\rho;\theta)$ in the "Multiagent Apprenticeship Learning" section expand to? Cannot seem to find a definition anywhere.
>
> **Response to Q1)**: $\psi$ is the cumulative regret and is defined as $\psi(\pi, \rho ; \theta) \doteq \sum_{i \in[n]} u_i\left(\rho_i, \pi_{-i} ; \pi\right)-u_i(\pi ; \mathbf{\theta})$. (See page 3, at the end of the paragraph starting with "One-shot games".) Note that this definition extends immediately to Markov games since any Markov game can be seen as a one-shot game in which the space of actions is taken to be the space of policies. This point is explained further in the sentences preceeding Corollary 1.
>
> >**Q2) ** $\bullet$ Assuming that Algorithm 3 was used on the spanish electricity market data, how was the observation distribution specified?
>
> **Response to Q2)**: In the electricity market experiments, the Markov game consists of an electricity seller who sets prices in the day ahead market and spot market, and $n$ buyers who demand electricity. We use price for the day ahead market, prices for the spot market, and aggregate demand (i.e., the sum of the demand across all buyers) of electricity for every hour from 2015 to 2019 as our observation space. The observation distribution then consists of the history distribution associated with this Markov game, which is pushed forward through a function that outputs sampled price trajectories and the sum of each buyer's individual demand.

---

> > ### Author Response · Authors · 2023-11-21
> >
> > Dear reviewer,
> >
> > Since the end of the discussion period is nearing, we were wondering whether you have further questions. We would like to have the opportunity to discuss and resolve potential further inquiries.
> >
> > Thanks in advance,
> > The authors

---

> > > ### Comment · Reviewer_WrBZ · 2023-11-22
> > > **Reviewer Response**
> > >
> > > Thank you for your response. Good to see that almost all weaknesses have been acknowledged. Though I am still a little torn on the proofs, they're just hard to follow for someone who does not have a core game theory background, given the paper's title does not mention game theory at all. However, my lack of expertise in verifying those convergence proofs cannot be ruled out, so take this with a grain of salt. A more experienced reviewer may have a different opinion.
> > >
> > > Also I hope you are aware that there is another paper with an almost identical title (https://arxiv.org/abs/1807.09936) which tackles the same problem albeit by extending single-agent GAIL to the multi-agent setting. Yours being a inverse game theoretic approach, the phrase "Generative-Adversarial" is a bit misleading to use for any min-max objective (another reviewer has also pointed this out).

---

> > > > ### Author Response · Authors · 2023-11-22
> > > >
> > > > Thank you for your quick reply!
> > > >
> > > > > Thank you for your response. Good to see that almost all weaknesses have been acknowledged. Though I am still a little torn on the proofs, they're just hard to follow for someone who does not have a core game theory background, given the paper's title does not mention game theory at all. However, my lack of expertise in verifying those convergence proofs cannot be ruled out, so take this with a grain of salt. A more experienced reviewer may have a different opinion.
> > > >
> > > > We tried to clarify the proofs as much as possible in the newly updated version of the paper. Please let us know if any confusions remain and we will seek to clarify it in the remaining day of discussion. We would also like to note that all our convergence results follow from known results in the optimization and reinforcement learning theory literature, and we have used no novel proof techniques that might cast shadow on the correctness of our results.
> > > >
> > > > > Also I hope you are aware that there is another paper with an almost identical title (https://arxiv.org/abs/1807.09936) which tackles the same problem albeit by extending single-agent GAIL to the multi-agent setting. Yours being a inverse game theoretic approach, the phrase "Generative-Adversarial" is a bit misleading to use for any min-max objective (another reviewer has also pointed this out).
> > > >
> > > > Regarding "Multi-Agent Generative Adversarial Imitation Learning" [1]. This work is cited in our paper, and a comparison of its results with regards to ours can be found in Table 1 of our paper. A subset of the same authors also have a similar follow-up paper called "Multi-agent adversarial inverse reinforcement learning" [2] for which we also have a comparison included in Table 1.
> > > >
> > > > Importantly, the focus of [1] is mostly behavioral cloning, and while [1] and [2] provide a mathematical characterization for inverse multiagent reinforcement learning (see for instance Section 3 of [1]), this is not a min-max but rather a non-convex single objective characterization (i.e., is not guaranteed to be solvable in polynomial-time) and does not concern Nash equilibria but rather quantal response equilibria in finite state and action Markov games. In sum, our results apply to a larger class of Markov games (i.e., beyond finite state and action) and equilibrium concepts (i.e. beyond quantal response), and provide a polynomial-time computation guarantee for a large class of Markov games (including finite state and action Markov games).
> > > >
> > > > All this said, we agree with you and reviewer F6LX regarding the use of the terminology in our "generative adversarial", and will replace this terminology with "min-max". We are also open to other suggestions.
> > > >
> > > > **References**
> > > >
> > > > [1] Song, Jiaming, et al. "Multi-agent generative adversarial imitation learning." Advances in neural information processing systems 31 (2018).
> > > >
> > > > [2] Yu, Lantao, Jiaming Song, and Stefano Ermon. "Multi-agent adversarial inverse reinforcement learning." International Conference on Machine Learning. PMLR, 2019.

---

### Author Response · Authors · 2023-11-17
**Common answer**

We thank all the reviewers for their time!

**Summary of contributions**: We provide a computationally efficient (i.e. polynomial-time) min-max optimization characterization of a large class of inverse equilibrium problems in inverse game theory and inverse multiagent learning. Our formulation provides a simple solution to a large number of problems, one of which dates back to the 1940s (i.e., the revealed preference problem [1]). We also generalize the apprenticeship learning paradigm to multiagent settings, characterize the solutions to multiagent apprenticeship learning, introduce a novel method to compute such solutions, and demonstrate the effectiveness of our approach at predicting prices in Spanish electricity markets.

**Common Reviewer Concern**: All reviewers expressed similar confusion regarding the whereabouts of the proofs of our theorems. We apologize for the confusion, which seems to have arisen from the fact that the detailed theorem statements in the appendix were renumbered, and as a result incorrectly referenced. Specifically, Theorems 3.2, 4.1, and 5.2 were restated in detail as Theorems 6.1, 6.2, and 6.3 respectively. We are uploading together with this rebuttal an updated appendix with corrected references and slightly expanded proofs (e.g., specifying omitted constants). We expect that this corrected appendix will address the reviewers' concerns.

**References**

[1] Paul A Samuelson. Consumption theory in terms of revealed preference. Economica, 15(60):243–253, 1948.

---

### Meta-Review · Area_Chair_b6MS · 2023-12-06

**Metareview:**

This paper studies inverse game theory, a generalization of inverse reinforcement learning to multi-agent settings, in which the parameters that make demonstrated behavior an equilibrium for a game are sought. Compared to previous work in this space, this paper considers a broader set of concave and continuous Markov games and the Nash equilibria solution concept while providing polynomial time guarantees. The approach is demonstrated experimentally using Spanish electicity data. The reviewers appreciated the contributions of the paper, but felt the proofs should be made more accessible. I also echo the reviewer opinion that the paper title should be changed to better distinguish it from "Multi-agent generative adversarial imitation learning," which differs only in the word order.

**Justification For Why Not Higher Score:**

Stronger experimental results with more comparisons would be needed to warrant a higher recommendation.

**Justification For Why Not Lower Score:**

The paper makes a solid technical contribution and the reviewers were uniformly positive about it (6+ ratings).

---

### Decision · Program_Chairs · 2024-01-16

Accept (spotlight)